# Neural incomplete factorization: learning preconditioners for the conjugate gradient method

**Paul Häusner**                                                              *paul.hausner@it.uu.se*
*Department of Information Technology, Uppsala University, Sweden*

**Ozan Öktem**                                                                      *ozan@kth.se*
*Department of Mathematics, KTH Royal Institute of Technology, Stockholm, Sweden*

**Jens Sjölund**                                                             *jens.sjolund@it.uu.se*
*Department of Information Technology, Uppsala University, Sweden*

**Reviewed on OpenReview:** *https://openreview.net/forum?id=FozLrZ3CI5*

## Abstract

The convergence of the conjugate gradient method for solving large-scale and sparse linear equation systems depends on the spectral properties of the system matrix, which can be improved by preconditioning. In this paper, we develop a computationally efficient data-driven approach to accelerate the generation of effective preconditioners. We, therefore, replace the typically hand-engineered preconditioners by the output of graph neural networks. Our method generates an incomplete factorization of the matrix and is, therefore, referred to as neural incomplete factorization (NeuralIF). Optimizing the condition number of the linear system directly is computationally infeasible. Instead, we utilize a stochastic approximation of the Frobenius loss which only requires matrix-vector multiplications for efficient training. At the core of our method is a novel message-passing block, inspired by sparse matrix theory, that aligns with the objective of finding a sparse factorization of the matrix. We evaluate our proposed method on both synthetic problem instances and on problems arising from the discretization of the Poisson equation on varying domains. Our experiments show that by using data-driven preconditioners within the conjugate gradient method we are able to speed up the convergence of the iterative procedure. The code is available at https://github.com/paulhausner/neural-incomplete-factorization.

## 1 Introduction

Solving large-scale systems of linear equations is a fundamental problem in computing science. Available solving techniques can be divided into *direct* and *iterative* methods. While direct methods, which rely on computing the inverse or factorizing the matrix, obtain accurate solutions they do not scale well for large-scale problems. Therefore, iterative methods, which repeatedly refine an initial guess to approach the exact solution, are used in practice when an approximation of the solution is sufficient (Golub & Van Loan, 2013).

The conjugate gradient method is a classical iterative method for solving equation systems of the form $Ax = b$ where the matrix $A$ is symmetric and positive definite. Further, the matrix is typically assumed to be sparse in the sense that it contains few non-zero elements (Carson et al., 2023). Problems of this form arise naturally in the discretization of elliptical PDEs, such as the Poisson equation, and a wide range of optimization problems, such as quadratic programs (Pearson & Pestana, 2020; Potra & Wright, 2000). The convergence speed of the method thereby depends on the spectral properties of the matrix $A$ (Golub & Van Loan, 2013). Therefore, it is common practice to improve the spectral properties of the equation system by preconditioning before solving it. Despite the critical importance of preconditioners for practical performance, it has proven notoriously difficult to find good designs for a general class of problems (Saad, 2003).

Recent advances in data-driven optimization combine classical optimization algorithms with data-driven methods. A common approach is replacing hand-crafted heuristics with parameterized functions such as neural networks that are trained against data (Banert et al., 2024; Chen et al., 2022). In this work, we combine this methodology with results known from sparse matrix theory by representing the matrix of the linear equation system as a graph that forms the input to a graph neural network. We then train the network on a set of training problems to predict suitable preconditioners.

To train our model in a computationally feasible way we are inspired by the incomplete factorization methods. These methods, which include the popular incomplete Cholesky factorization, compute a sparse approximation of the Cholesky factor of the matrix $\boldsymbol{A}$ that is applied to precondition the linear system (Golub & Van Loan, 2013; Scott & Tůma, 2023). However, incomplete factorization methods can suffer from breakdown, require significant time to compute, and are hard to parallelize on modern hardware (Benzi, 2002; Naumov, 2012). We overcome some of these limitations by introducing a scalable way to compute efficient preconditioners that do not suffer from breakdown. Our training scheme relies on matrix-vector multiplications only, which can be implemented efficiently and allows us to train the method on large-scale problems. Further, we show how to apply common heuristics to control the obtained sparsity pattern of the learned preconditioner to obtain flexible preconditioning matrices (Saad, 2003).

In contrast to the NeuralPCG model introduced by Li et al. (2023), our model utilizes insights from graph theory to motivate the underlying computational framework and align it with the training objective. This allows us to obtain more effective preconditioners – in the sense of reducing the number of iterations for the preconditioned system – while using a smaller model. Reducing the number of parameters, in turn, leads to a faster inference time which is critical to amortize the cost of training and applying the neural network. Furthermore, our method is fully self-supervised while previous data-driven preconditioners rely on the solution of the linear equation system during training which increases the time required to generate a suitable dataset significantly.

In summary, we introduce (i) a novel GNN architecture and (ii) an efficient way to compute the loss function, tailored to obtaining effective incomplete factorization preconditioners; finally, (iii) we show how to obtain flexible sparsity patterns of the output within this framework. Our data-driven method exhibits state-of-the-art results for data-driven preconditioners and performs on par with the baseline incomplete factorization methods.

## 2 Background

After introducing the conjugate gradient method, we briefly summarize the basics of graph neural networks which form the computational backend for our proposed method.

### 2.1 Conjugate gradient method

The conjugate gradient method (CG) is a well-established iterative method for solving symmetric and positive-definite (abbreviated spd and denoted by $S_n^{++}$) systems of linear equations of the form $\boldsymbol{A}\boldsymbol{x} = \boldsymbol{b}$. The algorithm does not require any matrix-matrix multiplications, making CG particularly effective when dealing with large-scale and sparse matrices (Saad, 2003). The method creates a sequence of search directions $\boldsymbol{p}_i$ from the corresponding Krylov subspace which are orthogonal with respect to the inner product induced by the matrix $\boldsymbol{A}$, i.e. $\boldsymbol{p}_i^\mathsf{T} \boldsymbol{A} \boldsymbol{p}_j = 0$ for $i \neq j$. In other words, the search directions are mutually *conjugate* to each other. These search directions are used to update the solution iterate $\boldsymbol{x}_k$. Since it is possible to compute the optimal step size in closed form for each search direction, the method is guaranteed to converge within $n$ steps (Shewchuk, 1994). Typically, the initial search direction $\boldsymbol{p}_0$ is chosen as the *gradient* of the corresponding quadratic program given by $\boldsymbol{b} - \boldsymbol{A}\boldsymbol{x}_0$ (Nazareth, 2009). In Algorithm 1 the preconditioned conjugate gradient method is shown. The original conjugate gradient algorithm can be recovered by setting $\boldsymbol{P} = \boldsymbol{I}$.

**Convergence**   The convergence of CG to the true solution $\boldsymbol{x}_\star$ depends on the spectral properties of the matrix $\boldsymbol{A}$. Using the condition number $\kappa(\boldsymbol{A})$, a linear worst-case bound of the error in the number of taken CG steps $k$ is given by (Carson et al., 2023)

$$\|\boldsymbol{x}_\star - \boldsymbol{x}_k\|_{\boldsymbol{A}} \leq 2 \|\boldsymbol{x}_\star - \boldsymbol{x}_0\|_{\boldsymbol{A}} \left( \frac{\sqrt{\kappa(\boldsymbol{A})} - 1}{\sqrt{\kappa(\boldsymbol{A})} + 1} \right)^k . \tag{1}$$

However, in practice the CG method often converges significantly faster and the convergence also depends on the distribution of eigenvalues and the initial residual. Clustered eigenvalues hereby lead to faster convergence. Further, a large condition number does not always imply a slow convergence of the iterative scheme. On the other hand, a small condition number leads to a fast convergence (Carson et al., 2023). A common approach to accelerate the convergence is to precondition the linear equation system to improve its spectral properties leading to faster convergence (Benzi, 2002).

**Preconditioning**  The underlying idea of preconditioning is to compute a cheap approximation of the inverse of $\boldsymbol{A}$ that is used to improve the convergence properties. A common way to achieve this is to approximate the matrix $\boldsymbol{P} \approx \boldsymbol{A}$ with an easily invertible preconditioning matrix $\boldsymbol{P}$, which allows us to compute the preconditioned search directions for each iteration in line 5 and line 10 of Algorithm 1 efficiently. This constraint can be achieved by constructing a (block) diagonal preconditioner or finding a (triangular) factorization (Benzi, 2002).

Finding a good preconditioner requires a trade-off between the time required to compute the preconditioner and the resulting speed-up in convergence (Golub & Van Loan, 2013). The Jacobi preconditioner simply approximates $\boldsymbol{A}$ with a diagonal matrix. The incomplete Cholesky (IC) preconditioner is a more advanced, but widely adopted, method. As the name suggests, the idea is to approximate the Cholesky decomposition of the matrix. The IC(0) preconditioner restricts the non-zero elements in the obtained triangular factor $\boldsymbol{L}$ to exactly the non-zero elements in the lower triangular part of $\boldsymbol{A}$. Thus, no fill-ins during the factorization are allowed. More general versions allow additional fill-ins of the matrix based on the position or the value of the matrix elements (Benzi, 2002) or allow flexible positions of non-zero elements such as as the modified incomplete Cholesky (MIC) preconditioner (Lin & Moré, 1999). The chosen amount of fill-in values determines how well the incomplete factorization approximates the original matrix and dictates how much the preconditioner accelerates the convergence of the iterative scheme. However, adding more fill-in elements increases the computational complexity of computing the incomplete factorization. Furthermore, it is desirable to know the amount of fill-in beforehand to avoid memory allocation problems (Scott & Tůma, 2023).

Finding new preconditioners is an active research area but is often done on a case-by-case basis. Newly developed methods are often tailor-made for specific problem classes and often do not generalize to new problem domains. General purpose preconditioners such as incomplete factorization methods and algebraic multigrid approaches are, nevertheless, popular and sufficiently effective for many problems in practice. Therefore, improvements upon these methods are of great interest for a broad community (Benzi, 2002; Pearson & Pestana, 2020).

**Stopping criterion**  In practice, due to numerical rounding errors, the residual is only approaching but never reaching zero. Further, the true solution $\boldsymbol{x}_\star$ is typically not available and therefore, equation (1) can

---

**Algorithm 1** Preconditioned conjugate gradient method (Nocedal & Wright, 1999)

---

1: **Input:** System of linear equations $\boldsymbol{A} \in S_n^{++}, \boldsymbol{b} \in \mathbb{R}^n$, Preconditioner $\boldsymbol{P} \approx \boldsymbol{A}, \boldsymbol{P} \in S_n^{++}$
2: **Output:** Solution to the linear equation system $\hat{\boldsymbol{x}}_\star$
3: Initialize starting guess $\boldsymbol{x}_0$
4: $\boldsymbol{r}_0 = \boldsymbol{b} - \boldsymbol{A}\boldsymbol{x}_0$
5: Solve $\boldsymbol{P}\boldsymbol{y}_0 = \boldsymbol{r}_0$ and set $\boldsymbol{p}_0 = \boldsymbol{y}_0$
6: **for** $k = 0, 1, \ldots,$ until convergence **do**
7: $\quad a_k = \langle \boldsymbol{r}_k, \boldsymbol{y}_k \rangle / \langle \boldsymbol{A}\boldsymbol{p}_k, \boldsymbol{p}_k \rangle$
8: $\quad \boldsymbol{x}_{k+1} = \boldsymbol{x}_k + a_k \boldsymbol{p}_k$
9: $\quad \boldsymbol{r}_{k+1} = \boldsymbol{r}_k - a_k \boldsymbol{A}\boldsymbol{p}_k$
10: $\quad$ Solve $\boldsymbol{P}\boldsymbol{y}_{k+1} = \boldsymbol{r}_{k+1}$
11: $\quad \beta_k = \langle \boldsymbol{r}_{k+1}, \boldsymbol{y}_{k+1} \rangle / \langle \boldsymbol{r}_k, \boldsymbol{y}_k \rangle$
12: $\quad \boldsymbol{p}_{k+1} = \boldsymbol{y}_{k+1} + \beta_k \boldsymbol{p}_k$
13: **return** $\boldsymbol{x}_k$

---

not be used as a stopping criterion for the algorithm either. Instead, the relative residual error $\|\boldsymbol{r}\|_2$ which is computed recursively in line 9 of Algorithm 1 is widely adopted (Shewchuk, 1994).

## 2.2 Graph neural networks

Graph neural networks (GNN) belong to an emerging family of neural network architectures well-suited to many real-world problems with a natural graph structure (Veličković, 2023). A (directed) graph $\mathcal{G} = (V, E)$ is a tuple consisting of a set of nodes $V$ and directed edges $E$ connecting two nodes in the graph $E \subseteq V \times V$. We assign every node $v \in V$ a node feature vector $\boldsymbol{x}_v \in \mathbb{R}^p$ and respectively every directed edge $e_{ij}$, connecting nodes $i$ and $j$, an edge feature vector $\boldsymbol{z}_{ij} \in \mathbb{R}^m$.

The widely adopted message-passing GNNs consist of multiple layers updating the node and edge feature vectors of the graph iteratively using permutation-invariant aggregations over the neighborhoods and learned update functions (Bronstein et al., 2021). Here, we follow the blueprint presented by Battaglia et al. (2018) to describe the update functions for a simple message-passing GNN layer. In each layer $l$ of the network the edge features are updated first by the network computing the features of the next layer $l + 1$ as

$$\boldsymbol{z}_{ij}^{(l+1)} = \phi_{\boldsymbol{\theta}_z^{(l)}}\left(\boldsymbol{z}_{ij}^{(l)}, \boldsymbol{x}_i^{(l)}, \boldsymbol{x}_j^{(l)}\right), \tag{2}$$

where $\phi$ is a parameterized function. The outputs of this function are also referred to as *messages*. Then, for each node $i \in V$ the features from its neighboring edges, in other words, the incoming messages, are aggregated using a suitable aggregation function. Typical choices include sum, mean and max aggregations. Any such permutation-invariant aggregation function is denoted here by $\oplus$. The aggregation of incoming messages over the neighborhood $\mathcal{N}$ of node $i$, which is defined as the set of adjacent nodes in the graph $\mathcal{N}(i) = \{j \mid (i, j) \in E\}$, is computed as

$$\boldsymbol{m}_i^{(l+1)} = \bigoplus_{j \in \mathcal{N}(i)} \boldsymbol{z}_{ji}^{(l+1)}. \tag{3}$$

Note that the neighborhood structure is typically kept fixed in GNNs and no edges are added or removed between the different layers. The final step of the message-passing GNN layer is updating the node features as

$$\boldsymbol{x}_i^{(l+1)} = \psi_{\boldsymbol{\theta}_x^{(l)}}\left(\boldsymbol{x}_i^{(l)}, \boldsymbol{m}_i^{(l+1)}\right). \tag{4}$$

The node and edge update functions $\phi$ and $\psi$ are typically parameterized using neural networks. The equations (2)–(4) describe the iterative scheme of message passing that is implemented by many popular GNNs. By choosing a permutation-invariant function in the neighborhood aggregation step (3) and since the update functions only act locally on the node and edge features, the learned function represented by the GNN itself is permutation equivariant and can handle inputs of varying sizes (Battaglia et al., 2018).

Many known algorithms share a common computational structure with this message-passing scheme making GNNs a natural parameterization for learned variants. This correspondence is typically referred to as algorithmic alignment in the literature (Dudzik & Veličković, 2022).

# 3 Method

In this section, we formulate the learning problem for a data-driven preconditioner and derive an efficient loss function. Then, we introduce a problem-tailored GNN architecture, and demonstrate how to extend the framework to yield flexible sparsity patterns and analyze its complexity.

**Learning problem**  Our final goal is to learn a mapping $f_{\boldsymbol{\theta}} : S_n^{++} \to S_n^{++}$ that takes an spd matrix $\boldsymbol{A}$ and predicts a suitable preconditioner $\boldsymbol{P}$ that improves the spectral properties of the system – and therefore also the convergence behavior of the conjugate gradient method.

In order to ensure convergence, the output of the learned mapping needs to be spd and is in practice often required to be sparse due to resource constraints. To ensure these properties, we restrict the mapping

to lower-triangular matrices with strictly positive elements on the diagonal denoted as $\mathcal{L}_n^+$ and enforce the same sparsity pattern as the input matrix. We then learn a mapping $\Lambda_{\boldsymbol{\theta}} : \mathcal{S}_n^{++} \to \mathcal{L}_n^+$. This can be achieved by using a suitable activation function and network architecture as discussed later. The matrices in $\mathcal{L}_n^+$ are guaranteed to be invertible and the computation thereof is computationally efficient since using forward-backward substitution requires only $\mathcal{O}(n^2)$ operations (Golub & Van Loan, 2013). For sparse matrices the triangular system can often be solved even more efficiently and scales with the number of non-zero elements (Davis et al., 2016). The parameterized function outputs a factorization of the preconditioning matrix that can be obtained via

$$\boldsymbol{P_\theta}(\boldsymbol{A}) = \Lambda_{\boldsymbol{\theta}}(\boldsymbol{A})\Lambda_{\boldsymbol{\theta}}(\boldsymbol{A})^\mathsf{T}, \tag{5}$$

which is a spd matrix if the diagonal elements of the triangular matrix are strictly positive i.e. $\Lambda_{\boldsymbol{\theta}}(\boldsymbol{A}) \in \mathcal{L}_n^+$ and can be easily inverted. To improve the convergence properties we aim to improve the condition number of the preconditioned system. However, computing the condition number scales very poorly for large problems as it requires $\mathcal{O}(n^3)$ floating-point operations. Therefore, we can not optimize the spectral properties of the matrix directly. Instead, we are inspired by incomplete factorization methods (Benzi, 2002). These methods try to find a sparse approximation of the Cholesky factor of the input matrix while remaining computationally tractable. In order to learn a good approximation model we assume matrices of interest are generated by a matrix-valued random variable $\mathbb{A}$ with an unknown distribution describing the underlying class of problems.

We train our learned preconditioner to approximately solve the matrix factorization optimization problem with additional sparsity constraints. The training objective for the data-driven model is given by

$$\hat{\boldsymbol{\theta}} \in \arg\min_{\boldsymbol{\theta}} \mathbb{E}_{\boldsymbol{A}\sim\mathbb{A}} \left[\|\Lambda_{\boldsymbol{\theta}}(\boldsymbol{A})\Lambda_{\boldsymbol{\theta}}(\boldsymbol{A})^\mathsf{T} - \boldsymbol{A}\|_F^2\right] \tag{6a}$$

$$\text{s.t. } \Lambda_{\boldsymbol{\theta}}(\boldsymbol{A})_{ij} = 0 \text{ if } \boldsymbol{A}_{ij} = 0, \Lambda_{\boldsymbol{\theta}}(\boldsymbol{A}) \in \mathcal{L}_n^+. \tag{6b}$$

Equation (6a) aims to minimize the distance between the learned factorization and the input matrix using the Frobenius norm distance. It is, however, also possible to use a different distance metric instead (Vemulapalli & Jacobs, 2015). When considering more advanced preconditioners, the no fill-in sparsity constraint (6b) can be relaxed slightly allowing more non-zero elements in the preconditioner. However, it is desirable that the required storage for the preconditioner is known beforehand and does not increase too much compared to the original system (Benzi, 2002).

**Scalable training** A practical problem with objective (6a) is that we cannot compute the expectation since we lack access to the distribution of $\mathbb{A}$. On the other hand, we have access to training data $\boldsymbol{A}_1, \boldsymbol{A}_2, \ldots, \boldsymbol{A}_n \sim \mathbb{A}$, so we consider the empirical counterpart of equation (6) – empirical risk minimization – where the intractable expected value is replaced by the sample mean which we can optimize using stochastic gradient descent to obtain an approximation of $\hat{\boldsymbol{\theta}}$.

To enhance computational efficiency we circumvent computing the matrix-matrix multiplication $\Lambda_{\boldsymbol{\theta}}(\boldsymbol{A})\Lambda_{\boldsymbol{\theta}}(\boldsymbol{A})^\mathsf{T}$ in the objective, by approximating the Frobenius norm using Hutchinson's trace estimator (Hutchinson, 1989)

$$\|\boldsymbol{B}\|_F^2 = \text{trace}(\boldsymbol{B}^\mathsf{T}\boldsymbol{B}) \approx \boldsymbol{w}^T\boldsymbol{B}^T\boldsymbol{B}\boldsymbol{w} = \|\boldsymbol{B}\boldsymbol{w}\|_2^2, \tag{7}$$

where the elements in the vector $\boldsymbol{w}$ are iid normal distributed random variables. By setting $\boldsymbol{B} = \Lambda_{\boldsymbol{\theta}}(\boldsymbol{A})\Lambda_{\boldsymbol{\theta}}(\boldsymbol{A})^\mathsf{T} - \boldsymbol{A}$ we obtain an unbiased estimator of the loss which requires only matrix-vector products (Martinsson & Tropp, 2020). The resulting objective is similar to the loss proposed by Li et al. (2023) but does not rely on computing the true solution to the problem beforehand, making it computationally more efficient as it avoids solving linear equation systems before the training. In Appendix D.2, we show that using this approximation as the loss function leads to similar results as training using the full Frobenius norm as the loss.

**Model architecture** Due to the strong connection of graph neural networks with matrices and numerical linear algebra (Moore et al., 2023), graph neural networks are a natural choice to parameterize the function $\Lambda_{\boldsymbol{\theta}}$. Furthermore, this allows us to enforce the sparsity constraints (6b) directly through the network architecture and avoids scalability issues arising in other network architectures since GNNs can exploit the sparse problem structure through the architecture directly.

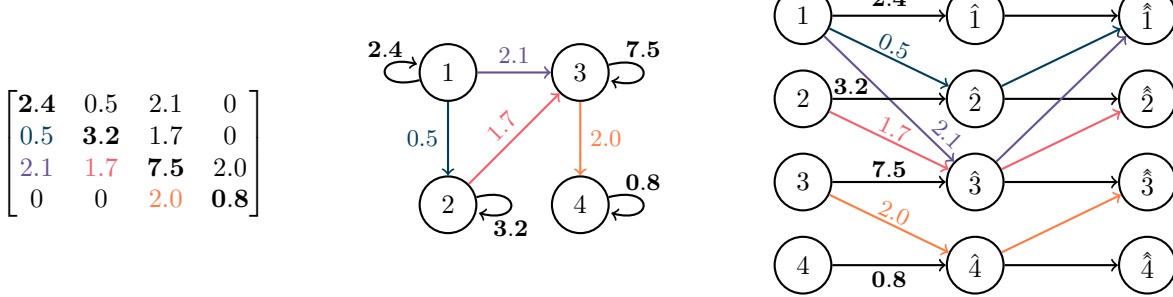

Figure 1: Different representations of the problem matrix $\boldsymbol{A}$. The classical linear algebra representation as a matrix (left). The Coates graph representation of the lower-triangular matrix used for the first message-passing step in each block (middle). The second step is executed on the Coates graph corresponding to the upper triangular part of the matrix, which can be obtained by flipping the edges of the lower triangular graph (not shown). The unrolled graph of the message passing resulting in a concatenation of the two directed graph representations used for the message passing in the graph neural network (right). Color is used to visualize edge and matrix element correspondence. Diagonal elements are in bold. The node labels in the graph indicate the corresponding row and column of the matrix.

On a high level, we interpret the matrix $\boldsymbol{A}$ of the input problem as the adjacency matrix of the corresponding graph. Thus, the nodes of the graph represent the columns/rows – the system is symmetric – of the input matrix and are augmented with an additional feature vector. The edges of the corresponding graph are the non-zero elements in the adjacency matrix. This transformation from a matrix to a graph is known as the Coates graph (Coates, 1959; Grementieri & Galeone, 2022). We use a total of eight node features related to sparsity structure and diagonal dominance of the matrix $\boldsymbol{A}$ (Cai & Wang, 2019; Tang et al., 2022). For the details about the network architecture we refer to Appendix C. There are two main issues with directly using the Coates graph of $\boldsymbol{A}$ for the message-passing scheme: Firstly, even though the input matrix is symmetric the latent representation of the edges – obtained after updating the input using the message passing scheme introduced in Section 2.2 – is generally not. Therefore, the matrix obtained after the message passing scheme is non-symmetric and requires additional care before being transformed into the lower triangular factorization. The second issue with directly using the Coates graph as input arises from the transformation of the final edge embedding of the GNN to a lower triangular matrix. This means some edges from the graph are removed based on the labeling of the nodes. In other words, all edges $e_{ij}$ such that $i \leq j$ are not included in the transformation of the network output. However, the GNN model that parameterizes the mapping $\Lambda_{\boldsymbol{\theta}}$ is permutation equivariant and, therefore, unable to capture this implicit dependency of node ordering and output transformation. To overcome these limitations and align our computational framework to the underlying objective of finding a sparse factorization, we introduce a novel GNN block that replaces the original Coates graph representation to execute the message-passing steps.

Instead of using the Coates graph representation of matrix $\boldsymbol{A}$ directly as the underlying graph for the message passing framework, we only use the graph corresponding to the lower triangular graph representation – that is the Coates graph of $tril(\boldsymbol{A})$ – to update node and edge features in the graph. This lower triangular structure directly corresponds to the sparsity pattern of the final output of our model. In other words, instead of first executing the message-passing steps and then transforming the output to a lower-triangular matrix, we reverse the order and first apply the lower-triangular transformation and then apply the GNN on the already transformed matrix. The corresponding Coates graph of the lower-triangular part of $\boldsymbol{A}$ used for message passing is depicted in the Figure 1.

However, only executing updates with the lower-triangular matrix prevents nodes with a lower label to obtain information from higher labeled nodes since the corresponding edges in the graph have been removed. Therefore, in a second step, the message-passing scheme is executed over the upper-triangular part of the matrix $\boldsymbol{A}$ instead. The edge features between the two consecutive layers are shared, in other words $\boldsymbol{z}_{ij}^{(n)} = \boldsymbol{z}_{ji}^{(n)}$.

After the two message-passing steps are computed, the original matrix $\boldsymbol{A}$ is used to augment the edge features by introducing skip connections. By changing the graph representation used for message passing, we explicitly encode the node ordering used in the downstream task into the network architecture. The detailed forward pass through the network architecture is described in Appendix C.

In order to highlight the connection between classical algorithms and our proposed network architecture, we consider the unrolled graph obtained from the GNN. This is shown in Figure 1 on the right-hand side. The unrolled graph is a König graph representation of the message-passing scheme (Doob, 1984). Here, each column of nodes represents a latent graph representation, starting with the initial graph on the left. Subsequent steps are obtained by applying the corresponding message-passing step over the edges of the lower and upper triangular part respectively. The additional dependency on the parameters $\boldsymbol{\theta}$ is omitted in this figure for clarity. The underlying motivation of the message-passing block introduced in the previous paragraph is that matrix multiplication can be represented in graph format by concatenating the graph representations of the two factors in König's graph representation. Each element in the product matrix can then be computed as the sum of the weights connecting the two corresponding nodes in the concatenated graphs. The skip connections are introduced as they represent an element-wise addition of the two matrices (Doob, 1984; Brualdi & Cvetkovic, 2008).

Our model consists of three blocks resulting in a total of six message-passing steps. To enforce the positive definiteness of the learned preconditioner the final diagonal elements of the output are transformed via

$$\Lambda_{\boldsymbol{\theta}}(\boldsymbol{A})_{ii} = \exp\left(\frac{1}{2} \cdot z_{ii}^{(N)}\right). \tag{8}$$

This activation function forces the diagonal elements to be strictly positive. Multiplying the final edge embedding by the constant term one-half before applying the non-linear activation avoids numerical problems and, based on our experiments, improves the convergence during training. Due to the connection to incomplete factorization methods, we refer to our model as neural incomplete factorization (NeuralIF).

**Additional fill-ins and droppings** Similar to the incomplete Cholesky method with additional fill-ins (Saad, 2003), processing on the graph can be applied to obtain both static and dynamic sparsity patterns for the learned preconditioner. Static level of fill-ins can be obtained by adding additional edges to the graph prior to the message passing during the symbolic phase. This corresponds to relaxing constraint (6b) to allow more non-zero elements. Adding all remaining edges to the graph is computationally intractable and has limited applicability to large-scale problems. Therefore, we use heuristics inspired by the level-based fill-ins for incomplete factorization methods, for example, by allowing all entries corresponding to the sparsity pattern of $\boldsymbol{A}^2$.

To obtain a dynamic sparsity pattern, we add a weighted $\ell_1$-penalty on the elements in the learned preconditioning matrix $\|\Lambda_{\boldsymbol{\theta}}(\boldsymbol{A})\|_1$ to the training objective (6a). This encourages the network to produce sparse outputs (Jenatton et al., 2011). During inference, we drop elements with a small magnitude to obtain an even sparser preconditioner. This can lead to faster overall solving times since the step to find the search direction in line 10 of Algorithm 1 scales with the number of non-zero elements in the preconditioner (Davis et al., 2016). Both of these approaches can also be combined to obtain a learned preconditioner similar to the dual threshold incomplete factorization (Saad, 1994).

By including additional edges, the forward pass of the GNN becomes computationally more expensive. Therefore, it is important to avoid increasing the non-zero elements too much to maintain the computational efficiency. Dropping additional elements, on the other hand, can be achieved with nearly no overhead during inference.

**Inference and complexity** When the trained model is used during inference to generate preconditioners, the output $\boldsymbol{P}_{\theta}(\boldsymbol{A}) = \Lambda_{\theta}(\boldsymbol{A})\Lambda_{\theta}(\boldsymbol{A})^{\mathsf{T}} \approx \boldsymbol{A}$ is used within Algorithm 1 to find the preconditioned search direction requiring only a single forward pass through the model.

For the complexity analysis, we assume that the number of non-zero elements in the sparse matrix $\boldsymbol{A}$ is on the order of the matrix size i.e. $nnz(\boldsymbol{A}) = \mathcal{O}(n)$, which is a common assumption when working with sparse matrices (Scott & Tůma, 2023). Further, the embedding size of the node and edge features in the hidden layers is constant and, therefore, omitted from the discussion below.

The space complexity of the data-driven preconditioner during inference is $\mathcal{O}(n)$ as only the non-zero elements ($\mathcal{O}(nnz)$) in the matrix $\boldsymbol{A}$ as well as the fixed-size node features ($\mathcal{O}(n)$) need to be stored. The time complexity of the forward pass requires $\mathcal{O}(nnz)$ operations for the edge update in equation (2) and $\mathcal{O}(n)$ steps for the node update given by equation (4). Here, a fixed-size neural network processes each node and edge representation individually. The aggregation step of the messages, given by equation (3), requires each node to aggregate the incoming edge features from all incident edges. Assuming every node has $d$ neighbors on average, this leads to a total complexity of $\mathcal{O}(n \cdot d) = \mathcal{O}(nnz)$ (Blakely et al., 2021). This operation can be computed efficiently using sparse matrix-vector multiplication or the scatter operation in PyTorch Geometric (Fey & Lenssen, 2019). Thus, the overall time complexity for the forward pass of the neural network preconditioner for a sparse matrix $\boldsymbol{A}$ is linear in the number of non-zero elements $\mathcal{O}(nnz)$.

The same linear time complexity is achieved by the classical incomplete Cholesky factorization without fill-ins. Going beyond the zero fill-in analysis conducted here is infeasible for general sparse matrices as it is highly dependent on the structure of the sparsity pattern (Ghai et al., 2019). In practice the neural network based implementation benefits from a higher parallelization ability leading to faster wall-clock computation times compared to highly optimized algorithms as shown in Section 4.2.

The implementation details can be found in Appendix C.

## 4 Results

The overall goal of preconditioning techniques is to reduce total computational time required to solve the linear equation system up to a given precision (here, we choose $10^{-6}$) measured by the residual norm as described in Section 3. The time used to compute the preconditioner beforehand, therefore, needs to be traded off with the achieved speed-up through the usage of the preconditioner. We compare the different methods based on both the time required to compute the preconditioner (P-time) and the time needed to solve the preconditioned linear equation system using the CG method (CG-time) which is related to the number of iterations required. However, the type and the sparsity of the obtained preconditioner also influences the time-per-iteration. For instance in factorized preconditioners the sparse triangular solve method scales with the number of non-zero elements (Davis et al., 2016) while diagonal preconditioners can be applied in a fully parallelized fashion. The implementation details for the conjugate gradient method and different baselines and methods are described in Appendix C.

We consider two different datasets in our experiments. The first dataset consists of synthetically generated problems where we can easily control the size and sparsity of the generated problem instances. The other class is motivated by problems arising in scientific computing by discretizing the Poisson PDE on varying grids using the finite element method. The details for the dataset generation can be found in Appendix A.

For numerical experiments we use a single NVIDIA-Titan Xp with 12 GB memory. For baseline preconditioners, which are not able to be accelerated directly using GPUs, we use 6 Intel Core i7-6850K @ 3.60 GHz processors for the computations. The (preconditioned) conjugate gradient method is always run on the CPU to ensure a fair comparison between the performance of the preconditioners. Both of our models as well as the data-driven NeuralPCG baseline are trained for a total of 50 epochs. However, convergence can usually be observed significantly earlier. For the synthetic dataset we use a batch size of 5, while for the problems arising from the PDE discretization we only use a batch size of 1 due to resource constraints. This leads to a total training time of 40 minutes for the synthetic dataset and 55 minutes for the PDE dataset for each model. For all training schemes we utilize early stopping based on the validation set performance as an additional regularization measure. However, training can be further accelerated by applying different validation strategies and stopping training once convergence is observed.

Further acceleration of our neural network-based preconditioner can be obtained by batching the problem instances. This allows parallelization of the computation for the preconditioners leading to a smaller precomputation overhead. However, to ensure a fair comparison in our experiments, we handle each problem individually in our experiments.

### 4.1 Synthetic problems

The results and statistics about the preconditioned systems from the experiments using the synthetic dataset of size $n = 10\,000$ with 1% non-zero elements are shown in Table 1. As a baseline, we include the standard CG implementation without any additional preconditioner. We can see that our learned preconditioner is significantly faster to compute while maintaining similar performance in terms of reduction in number of iterations as the incomplete Cholesky (IC) method without additional fill-ins. This makes our approach, in terms of total solving time, on average $\sim 25\%$ faster than incomplete Cholesky.

Compared to the data-driven NeuralPCG (Li et al., 2023) method, our method is faster to compute during inference which leads to a smaller P-time, since the number of parameters is significantly smaller. Further, we are able to reduce the number of required iterations to solve the problem using preconditioned CG significantly more. In summary, our data-driven approach shows the ability of the learned preconditioner to accelerate the solving procedure both compared to classical methods and other data-driven preconditioning techniques.

**Dynamic sparsity pattern**   The results for the learned preconditioner with additional sparsity, through dropping elements by value as described in Section 3, are shown as NeuralIF-sp. Even though the output of the model contains only around half of the elements compared to the methods without fill-ins, the performance in terms of iterations does not suffer significantly. This can be exploited within the forward-backward solves which scale with the number of non-zero elements. Therefore, the overall time to solve the system is on average shortest using the sparsified NeuralIF preconditioner across all tested preconditioning methods.

**Solving times**   The distribution of total solving times is shown in Figure 2, where each point represents a single test problem instance that is solved with each respective method. The time shown includes both the time required to compute the preconditioner (P-time) and the time to run the preconditioned CG method (CG-time). Overall, the variance of solving time between different problems is small for all considered methods which is due to the fact that the generated problems are very similar. The Jacobi and incomplete Cholesky preconditioner are effective for most problems. In comparison, the NeuralPCG method is not able to speed up the computational time compared to the other preconditioning baselines. We can see that both of our NeuralIF preconditioners outperform the other preconditioning methods on nearly all problem instances but similar to other methods exhibit slightly worse performance on a few problem instances.

**Eigenvalue distribution**   In Figure 3, the ordered eigenvalues of the preconditioned linear equation system for one test problem are shown. Here, we can see that NeuralIF is especially able to reduce large eigenvalues compared to all the other preconditioners. However, compared to incomplete Cholesky the smaller eigenvalues of the NeuralIF preconditioned system decrease earlier and the smallest eigenvalue of the system is slightly smaller. This results in a slightly worse convergence behavior in the limit of our data-driven method.

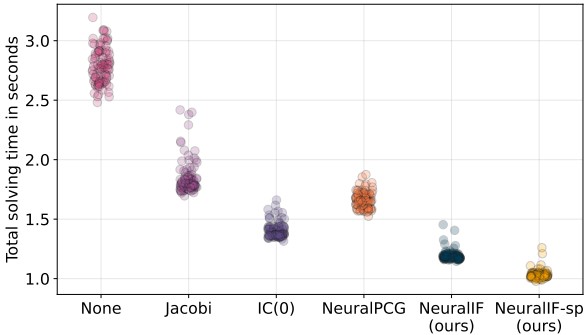

Figure 2: Total solving time for each test problem instance from the synthetic dataset.

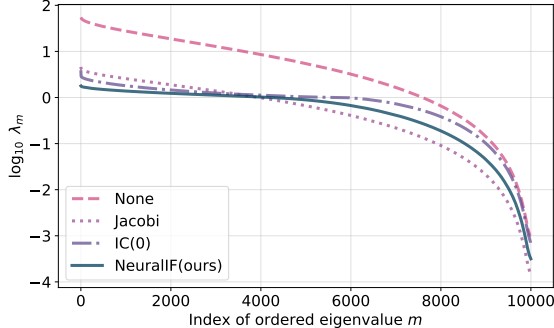

Figure 3: Ordered eigenvalues of the preconditioned linear equation system in log-scale.

Table 1: Mean results for the synthetic dataset for $n = 10\,000$ with 1% non-zero elements on 100 test instances. The first column shows the condition number of the preconditioned system $\boldsymbol{L}^{-1}\boldsymbol{A}\boldsymbol{L}^{-\top}$. The second column lists the preconditioner's sparsity. Remaining columns list performance-related figures for the preconditioned conjugate gradient method: the third column lists the computation time for the preconditioner (P-time), the fourth lists the time for finding the solution and the number of iterations required running the preconditioned conjugate gradient method (CG-time/Iters.) and the fifth column shows the combined time for computing the preconditioner and solving the system. All times are in seconds.

| Preconditioner | Cond. number $\kappa \downarrow$ | Sparsity $\uparrow$ | P-time $\downarrow$ | CG-time (its.) $\downarrow$ | Total time $\downarrow$ |
|---|---|---|---|---|---|
| None | $60\,834.17$ | - | - | 2.79 (935.99) | 2.79 |
| Jacobi | $33\,428.86$ | **99.99%** | **0.005** | 1.83 (689.82) | 1.84 |
| IC(0) | **$4\,707.17$** | 99.49% | 0.247 | 1.16 (**260.64**) | 1.40 |
| NeuralPCG | $7\,240.72$ | 99.49% | 0.123 | 1.54 (318.86) | 1.66 |
| **NeuralIF (ours)** | $4\,921.76$ | 99.49% | 0.028 | 1.16 (267.08) | 1.19 |
| **NeuralIF-sp (ours)** | $5\,581.44$ | 99.77% | 0.021 | **1.01** (286.02) | **1.03** |

**Breakdown** One inherent limitation of the incomplete Cholesky method is that it suffers from breakdown in 4% of the synthetic test problems due to the numerical instabilities of the method (Benzi, 2002). These instances are not included in solving time in Table 1 but are very costly in practice since it requires a restart of the solving procedure. In contrast, our data-driven approach consistently generates a suitable preconditioner.

## 4.2 Poisson PDE problems

In the second problem, we are focusing on problems arising from the discretization of Poisson PDEs using the finite element method. In order to train our learned preconditioner efficiently, we create a subset of small training problems with matrices of size between $20\,000$ and $100\,000$ with up to $500\,000$ non-zero elements. The results for these problems are summarized in Table 2. Here, MIC is the modified incomplete Cholesky method with the same number of non-zero elements as IC(0) but potentially different locations of the non-zero elements which is determined dynamically (Lin & Moré, 1999). Allowing a more flexible sparsity pattern can lead to better results but additional pre-computation time is required to determine the position of non-zero elements. Among all preconditioners without fill-ins, our data-driven method performs best as it is very efficient to compute and reduces the required iterations vastly.

**Additional fill-ins** Among the preconditioners with additional fill-ins, the MIC+ preconditioner further allows additional elements based on the number of non-zero elements in each row of the matrix which improves the preconditioner at the cost of a more expensive pre-computation time. Our NeuralIF(1) preconditioner is obtained by allowing non-zero elements in the non-zero locations of the matrix $\boldsymbol{A}^2$ which is a widely-adopted

Table 2: Results for the small Poisson PDE problems from the training distribution using a subset of the columns shown in Table 1. The first set of method does not use fill-ins while in the second part additional non-zeros are added based on the position and value of the non-zero elements.

| Preconditioner | P-time | CG-time (its.) | Total time |
|---|---|---|---|
| None | - | 1.72 (856.38) | 1.72 |
| Jacobi | **0.004** | 1.01 (466.85) | 1.02 |
| IC(0) | 0.022 | **0.51** (**166.05**) | 0.53 |
| MIC | 0.029 | 0.56 (180.94) | 0.59 |
| NeuralPCG | 0.018 | 0.57 (189.97) | 0.59 |
| **NeuralIF(0) (ours)** | 0.014 | **0.51** (168.09) | **0.52** |
| MIC+ | 0.044 | **0.39** (**100.95**) | **0.43** |
| **NeuralIF(1) (ours)** | **0.030** | 0.44 (111.37) | 0.47 |

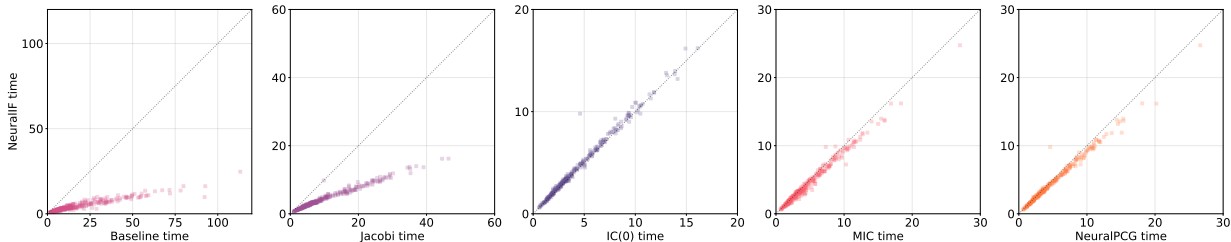

Figure 4: Pairwise comparison of total solving times (computation time of the precondition and solving time of the preconditioned linear system) of the NeuralIF preconditioner with the other preconditioners without fill-ins on the 300 large PDE problem instances. Instances towards the lower right part of the plot indicate that our method is faster, otherwise the baseline. Note that there are different axis scales for each comparison.

heuristic (Chow, 2000). Here, computing the static sparsity pattern before the message passing requires a significant amount of time and only minimal performance optimization of our algorithm is implemented. Therefore, the highly optimized implementation of the modified incomplete Cholesky method performs slightly better but the learned preconditioner improves significantly due to the added fill-ins compare to the previous methods without fill-ins.

**Generalization to larger problems**   Since real-world problems are often significantly larger than the training instances, we evaluate our methods also on problem instances of sizes up to 500 000 containing 3 000 000 non-zero elements. Since the problems in the dataset containing large instances are more diverse in terms of size, number of non-zero elements, and difficulty in solving them compared to the training dataset, we show a pairwise comparison in Figure 4. Here, we compare the total time it takes to solve the system using NeuralIF to solving the problem using various other preconditioners. We can see that for the larger problem instances, the NeuralIF preconditioner performs on par with the IC(0) method and even outperforms MIC, even though the problems are significantly larger than the training instances showing the generalization abilities of our method on problems very different from the training domain.

We additionally compare the scaling of NeuralIF with the incomplete Cholesky method in Figure 5. In the plot, we compare the number of non-zero matrix elements – which strongly correlates with the size of the

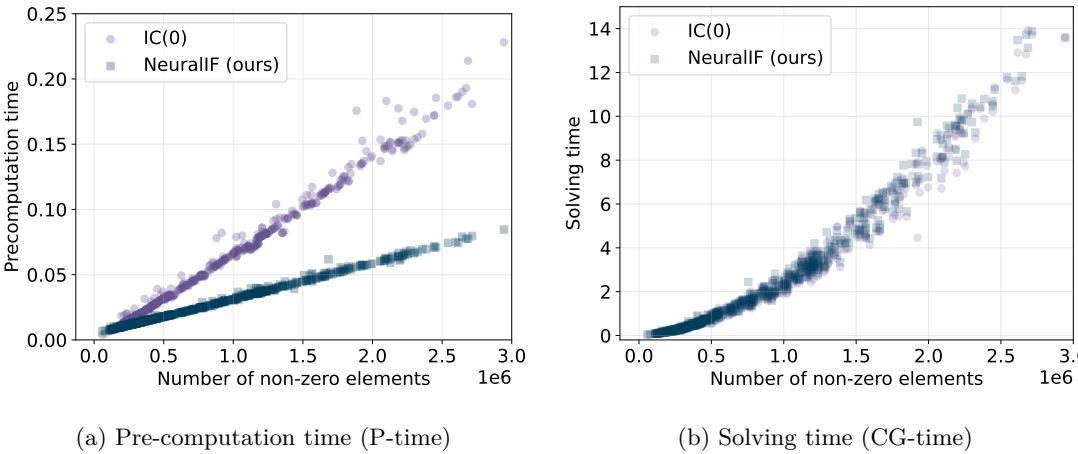

(a) Pre-computation time (P-time)          (b) Solving time (CG-time)

Figure 5: Comparison of the computational time required for the incomplete Cholesky and NeuralIF preconditioner with respect to the matrix size measure in number of non-zero elements on both the instances from the training distribution and problem instances outside of the training domain. We are using 600 problem instances from the generated Poisson PDE problems. The generated outputs have by construction the same number of non-zero elements as the input matrix.

matrix – with the time that is required to compute the preconditioner (P-time, Figure 5a) as well as the time required to run the preconditioned conjugate gradient method (Figure 5b).

The results show that our model exhibits a very good size generalization as it is able to produce efficient preconditioners even for matrices which are a magnitude of size bigger than the training instances. Further, we can see that the learned preconditioner scales better in terms of P-time for larger matrices than the incomplete Cholesky method and the variance for the preconditioner computation is decreased.

Additional results for both datasets can be found in Appendix D.

## 5 Discussion & Limitations

Here, we discuss several limitations and potential improvements with our proposed framework. However, we emphasize that many of these problems affect all learning-to-optimize methods and are not specific to our learned preconditioner.

**Training**   To train data-driven preconditioners, we assume that there is a sufficiently large set of problem instances from of a distribution $\mathbb{A}$ that share some similarity. In practice, not all problem domains give rise to such a distribution. Further, the time invested in the model training needs to be amortized over the speedup obtained during the inference phase of the learned preconditioner. However, this model training replaces the otherwise time-consuming manual tuning of preconditioners and is not a specific issue with our method but a limitation of the general learning-to-optimize framework (Chen et al., 2022). This also motivates following our self-supervised learning approach since it allows us to train the model on unsolved problem instances. The trained model can then, in principle, be applied to accelerate solving the training instances making it easier to amortize the model training.

**Convergence**   While the method works well in our numerical experiments and general convergence is ensured, no guarantees about the convergence speed can be made and it is likely that for some problem classes the usage of the learned preconditioner leads to an increase in overall solving time. That said, as seen in the experiments also classical preconditioners suffer from this problem and our method avoids pitfalls such as breakdowns in incomplete Cholesky (Benzi, 2002).

**Future work**   A promising future research direction is to learn more flexible sparsity patterns used for preconditioners alongside the values to fill in. This can be achieved by changing the graph structure used for message passing but requires additional care due to the combinatorial nature of the problem. Further, the incomplete factorization loss used to train our model is only a heuristic and not directly related to the complex convergence behavior of the conjugate gradient method. Extending the approach to directly take into account the downstream task instead of relying on the heuristic factorization approach has the potential to further improve the learned preconditioner and lead to faster convergence (Bansal et al., 2023).

## 6 Related work

Data-driven optimization or "learning-to-optimize (L2O)" is an emerging field aiming to accelerate numerical optimization methods by combining them with data-driven techniques (Amos, 2023; Chen et al., 2022). For example, a neural network can be trained to directly predict the solution to an optimization problem (Grementieri & Galeone, 2022) or replace some typically hand-crafted heuristic within a known optimization algorithm (Bengio et al., 2021).

**L2O using GNNs**   Graph neural networks have been recognized as a suitable computational backend for problems in data-driven optimization and linear algebra (Moore et al., 2023). Chen et al. (2023) study the expressiveness of GNNs with respect to their power to represent linear programs on a theoretical level. Using the Coates graph representation, Grementieri & Galeone (2022) develop a sparse linear solver for linear equation systems. They represent the linear equation system as the input to a graph neural network which is trained to approximate the solution to the equation system directly. However, no guarantees for the solution

can be obtained. In contrast, our method benefits from the convergence properties of the preconditioned conjugate gradient method.

Following an AutoML approach, Tang et al. (2022) instead try to predict a good combination of solver and preconditioner from a predefined list of techniques using supervised training. The NeuralIF preconditioner instead focuses on further accelerating the existing CG method and does not need any explicit supervision during training. Sjölund & Bånkestad (2022) use the König graph representation instead to accelerate low-rank matrix factorization algorithms by representing the matrix multiplication as a concatenation graph similar to our approach. However, their architecture utilizes a graph transformer while our approach works in a fully sparse setting. This avoids the scalability issues of transformer architectures and makes it better suited for large-scale problems.

**Data-driven CG**   Incorporating deep learning into the conjugate gradient method has been utilized in several ways previously. Kaneda et al. (2023) suggest replacing the search direction in the conjugate gradient method with the output of a neural network. This approach can, however, not be integrated with existing solutions for accelerating the CG method and does not allow further improvements through preconditioning. Furthermore, to ensure convergence all previous search directions need to be saved and the full Gram–Schmidt orthonormalization needs to be computed in every iteration making it prohibitively expensive.

There have also been some earlier approaches to learning preconditioners for the conjugate gradient method following similar ideas as our proposed NeuralIF method. Ackmann et al. (2021) use a fully-connected neural network to predict the preconditioner for climate modeling using a supervised loss. Sappl et al. (2019) use a convolutional neural network (CNN) to learn a preconditioner for applications in water engineering by optimizing the condition number directly. However, both of these approaches are only able to handle small-scale problems and their architecture and training is limited due to their poor scalability compared to our suggested approach. Utilizing CNN architectures, predicting sparsity patterns for specific types of preconditioners (Götz & Anzt, 2018) and general incomplete factorizations (Stanaityte, 2020) has also been suggested previously. In comparison, our learned method is far more general and can be applied to a significantly larger class of problems. Even though CNNs are widely adopted in the previous approaches, they are not well-suited to represent the underlying problem given by a matrix in contrast to GNNs. More recently, learned preconditioners have also been developed in the more general context of the GMRES solver for non-symmetric and indefinite matrices using graph neural networks (Häusner et al., 2024).

**Preconditioning**   Incomplete factorization methods are a large area of research. In most cases, the aim is to reduce computational time and memory requirements by ignoring elements based on either position or value. However, if done too aggressively, the method may break down due to numerical issues and require expensive restarts. Perturbation and pivoting techniques can mitigate but not completely eliminate such problems (Scott & Tůma, 2023). In contrast, our method does not suffer from breakdown as the positive-definiteness of the output is ensured a posteriori. Numerical efficiency can also be improved by reordering to reduce fill-in (for IC($\ell$) with $\ell > 0$) or by finding rows of the matrix that can be eliminated in parallel (Gonzaga de Oliveira et al., 2018).

Chow & Patel (2015) formulate incomplete factorization as the problem of finding a factorization that is exact on the given sparsity pattern as a feasibility problem that can be solved approximately. In contrast, our method only approximates the Frobenius norm minimization in equation (6) without enforcing constraints, which reduces pre-computation times. However, both approaches are quite similar in the sense that both methods aim to minimize the difference of the sparse factorization and the original matrix. While our method is using a fully-amortized learned optimization approach to minimize the full Frobenius norm distance (Amos, 2023), Chow & Patel (2015) directly optimize the residuals of non-zero elements in the approximate factorization. The latter approach also allows further acceleration using methods from distributed computing (Anzt et al., 2018) but typically requires highly optimized implementations to achieve competitive results.

For problems arising from elliptical PDEs, multigrid preconditioning techniques have shown to be very effective. These type of methods utilize the underlying geometric structure of the problem to obtain a preconditioner (Pearson & Pestana, 2020). Multigrid methods have also been successfully combined with deep learning techniques (Azulay & Treister, 2022). A potential drawback of this type of preconditioner is,

however, that the technique is computationally quite expensive compared to other methods and the resulting preconditioner is typically non-sparse.

## 7    Conclusions

In this paper we introduce NeuralIF, a novel and computationally efficient data-driven preconditioner for the conjugate gradient method to accelerate solving large-scale and sparse linear equation systems. Our method is trained to predict the sparse Cholesky factor of the input matrix following the widespread idea of incomplete factorization preconditioners (Benzi, 2002). To obtain a computationally efficient loss function, we derive a stochastic approximation of the Froebnius loss based on Hutchinson's trace estimator (Hutchinson, 1989). This allows us to train our model using matrix-vector multiplications only which can be efficiently implemented for the large-sclae sparse matrices.

We use the problem matrix $A$ as an input to a graph neural network, which processes the input and produces the desired approximate factorization. The network architecture aligns with the objective to minimize the distance between the learned output and the input matrix $A$ based on insights from graph theory using the Frobenius norm as a distance measure. Our experiments show that the proposed method is competitive against general-purpose preconditioners both on synthetic and real-world problems and allows the creation of both dynamic and static sparsity patterns. Our work shows the large potential of data-driven techniques in combination with insights from numerical optimization and the usefulness of graph neural networks as a natural computational backend for problems arising in computational linear algebra (Moore et al., 2023).

## Acknowledgments

We would like to thank Daniel Gedon, Fredrik K. Gustafsson, Sebastian Mair and Peter Munch for their feedback and discussions. This work was supported by the Wallenberg AI, Autonomous Systems and Software Program (WASP) funded by the Knut and Alice Wallenberg Foundation. The computations were enabled by the Berzelius resource provided by the Knut and Alice Wallenberg Foundation at the National Supercomputer Centre.

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

# A   Dataset details

The (preconditioned) conjugate gradient method is a standard iterative method for solving large-scale systems of linear equations of the form

$$\boldsymbol{A}\boldsymbol{x} = \boldsymbol{b} \tag{9}$$

where $\boldsymbol{A}$ is symmetric and positive definite (spd) i.e. $\boldsymbol{A} = \boldsymbol{A}^\mathsf{T}$ and $\boldsymbol{x}^\mathsf{T}\boldsymbol{A}\boldsymbol{x} > 0$ for all $\boldsymbol{x} \neq 0$. We also write $\boldsymbol{A} \in S_n^{++}$ to indicate that $\boldsymbol{A}$ is a $n \times n$ spd matrix (Golub & Van Loan, 2013). The method is especially efficient when the matrix $\boldsymbol{A}$ is large-scale and sparse since the required matrix-vector products shown in Algorithm 1 can efficiently exploit the structure of the matrix in this case (Saad, 2003).

In this section, we specify different problem settings leading to distributions $\mathbb{A}$ over matrices of the desired form such that the conjugate gradient method is a natural choice for solving the problem. Given a distribution we accessing via a number of samples, our goal is to train a model using the empirical risk minimization objective shown in the main paper to compute an incomplete factorization of a given input and utilize the resulting output as an effective preconditioner in the CG algorithm.

In total, we are testing our method on two different datasets but vary the parameters used to generate these datasets. The first problem dataset considers synthetic problem instances. The other test problem arises in scientific computing where large-scale spd linear equation systems can be obtained naturally from the discretization of elliptical PDEs.

The size of the matrices considered in the experiments is between $n = 10\,000$ and $n = 500\,000$ for all datasets in use. Throughout the paper we assume that the problem at hand is (very) sparse and spd. Thus, the preconditioned conjugate gradient method is a natural choice for all these problems. We ensure that the matrices are different by using unique and non-overlapping seeds for the problem generation. The datasets are summarized in Table 3. For the graph representation used in the learned preconditioner, the matrix size corresponds to the number of nodes in the graph and the number of non-zero elements corresponds to the number of edges connecting the nodes. In the following, the details for the problem generation are explained for each of the datasets.

Table 3: Summary of the datasets used with some additional statistics on size of the matrices and the number of corresponding non-zero elements. Samples refer to number of generated problems in the train, validation and test set respectively.

| Dataset | Samples | Matrix size | Non-zero elements (nnz) | Sparsity |
|---|---|---|---|---|
| Synthetic | $1\,000/10/100$ | $10\,000$ | $\sim 1\,000\,000$ | $99\%$ |
| Poisson - train | $750/15/300$ | $20\,000 - 150\,000$ | $800\,000 - 500\,000$ | $> 99.9\%$ |
| Poisson - test | $-/-/300$ | $100\,000 - 500\,000$ | $500\,000 - 3\,000\,000$ | $> 99.9\%$ |

## A.1   Synthetic problem

The set of random test matrices is constructed by choosing a sparsity parameter $p$ which indicates the expected percentage of non-zero elements in the matrix. It is also possible to specify $p$ itself as a distribution. We choose the non-zero probability such that the resulting spd matrices which are generated with equation equation (10) have around 99% total sparsity (therefore the generated problems have 1 million non-zero elements). We then create the problem by sampling a matrix $\boldsymbol{A}$ with $p(\boldsymbol{A}_{ij} = 0) = p$ and $p(\boldsymbol{A}_{ij}|\boldsymbol{A}_{ij} \neq 0) \sim \mathcal{N}(0, 1)$. In order to ensure the final matrix is spd we compute the test sample as

$$\boldsymbol{M} = \boldsymbol{A}\boldsymbol{A}^\mathsf{T} + \alpha \boldsymbol{I} \tag{10}$$

where $\alpha \approx 10^{-3}$ is used to ensure the resulting problem is positive definite. The right hand side of the linear equation system is sampled uniformly $\boldsymbol{b} \sim \mathcal{U}[0, 1]^n$.

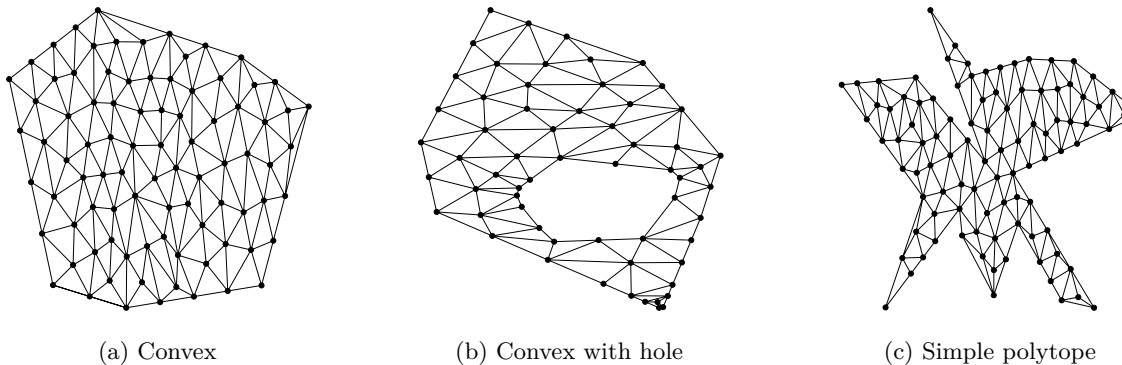

(a) Convex      (b) Convex with hole      (c) Simple polytope

Figure 6: Coarse sample meshes for each of the three different domain distributions used to generate training data for the 2-dimensional Poisson PDE problem.

Problems of this form also arise when minimizing quadratic programs since the equation equation (9) represents the first-order optimality condition for an unconstrained QP (Stellato et al., 2020). Thus, the problem represented here can be encountered in various settings making it an interesting case study.

### A.2 Poisson equation

The Poisson equation is an elliptical partial differential equation (PDE) and one of the most fundamental problems in numerical computational science (Langtangen & Logg, 2016). The problem is stated as solving the boundary value problem

$$-\nabla^2 u(x) = f(x) \qquad x \in \Omega \tag{11a}$$

$$u(x) = u_D(x) \quad x \in \partial\Omega \tag{11b}$$

where $f(x)$ is the source function and $u = u(x)$ is the unknown function to be solved for. By discretizing this problem using the finite element method and writing it in matrix form a system of linear equations of the form $\boldsymbol{Lx} = \boldsymbol{b}$ is obtained where the stiffness matrix $\boldsymbol{L}$ is sparse and spd. For large $n$, this problem therefore can be efficiently solved using the conjugate gradient method given a good preconditioner.

In order to obtain a distribution $\mathbb{A}$ over a large number of Poisson equation PDE problems, we consider a set of two-dimensional meshes that are generated by sampling from a normal distribution and creating a mesh based on the sampled points. We consider three different cases for the mesh generation: (a) we form the convex hull of the sampled points and triangulate the resulting shape, (b) we sample two sets of points and transform them into convex shapes as before. The second samples have a significantly smaller variance. The final shape that is used for triangulation is then obtained as the difference between the two samples. Finally, (c) we generate a simple polytope from the sampled points. An example for each of the generated shapes is shown in Figure 6.

The sampled meshes are discretized using triangular basis elements with the finite-element method and refined using the scikit-fem python library (Gustafsson & McBain, 2020). Due to the discretization based on the provided mesh the size of the generated matrix is variable. Based on the stiffness matrices arising from the discretized problems the NeuralIF preconditioner is trained.

## B Connection to sparse approximate inverse preconditioners

Instead of modeling the preconditioner as a sparse approximation of the factorization of $\boldsymbol{A}$, as in the incomplete factorization setting, sparse approximate inverse preconditioners (SPAI) directly approximate the inverse of $\boldsymbol{A}$ (Scott & Tůma, 2023). In other words, the goal of SPAI preconditioners is finding a matrix $\boldsymbol{M}$ such that $\boldsymbol{M} \approx \boldsymbol{A}^{-1}$ subject to sparsity constraints. Similar to our objective function in equation (6a), the objective of

SPAI preconditioners is usually also modeled as a Frobenius norm minimization problem

$$\min_{M \in \mathcal{S}} \|\boldsymbol{I} - \boldsymbol{M}\boldsymbol{A}\|_F \tag{12}$$

where $\boldsymbol{M}$ is restricted to have a predetermined sparsity pattern $\mathcal{S}$ (Benzi & Tuma, 1999). The solution to this problem can be computed efficiently but the output is neither guaranteed to be symmetric nor positive definite. Instead, the factorization of $\boldsymbol{M}$ can be obtained by reformulating the problem leading to the class of factorized sparse approximate preconditioners. In this class of methods, the distance between the factorized preconditioner and the unknown Cholesky factor of $\boldsymbol{A}^{-1}$ is computed (Benzi et al., 1996; Scott & Tůma, 2023).

Learning the sparse inverse approximation of a matrix directly from data and applying the learned function as a preconditioner is an interesting direction for future research (Bånkestad et al., 2024).

## C  Implementation details

In the following, we describe the details of the network architecture our NeuralIF model uses as well as the details of the model training, testing, and inference. Our method is implemented using PyTorch (Paszke et al., 2019) and PyTorch Geometric (Fey & Lenssen, 2019).

### C.1  Model architecture

Here, we summarize in detail the architecture choices for our NeuralIF preconditioner and provide details on the implementation and training of the method. In total, our model consists of $1\,780$ learnable parameters which is significantly less than previous approaches and allows for a fast inference time.

**Node and edge features**  In total eight features are used for each node thus $\boldsymbol{x}_i \in \mathbb{R}^8$. The first five node features listed in Table 4 are implemented following the *local degree profile* introduced by Cai & Wang (2019). The dominance and decay features shown in the table are originally introduced by Tang et al. (2022). Additionally, we use the position of the node in the matrix as the final input feature.

As edge feature, only the scalar value of the non-zero matrix entries are used. In deeper layers of the network, when skip connections are introduced, the edges are augmented with the original matrix entries. This effectively leads to having two edge features in these layers: the computed edge embedding of the current layer $\boldsymbol{z}_{ij}^{(l)}$ as well as the original matrix entry $a_{ij}$. However, from these only a one-dimensional output is computed $\boldsymbol{z}_{ij}^{(l+1)}$ as described in the following.

**Message passing block**  At the core of our method, we introduce a novel message passing block which aligns with the objective of our training. The underlying idea is that matrix multiplication can be represented in graph form by concatenating the two Coates graph representations as described previously (Doob, 1984). The pseudocode for the message passing scheme is shown in Algorithm 2. The message passing block consists of two GNN layers as introduced in Section 3 which are concatenated to mimic the matrix multiplication. To

Table 4: List of node features used in the graph neural network.

| Feature name | Description |
| --- | --- |
| deg(v) | Degree of node $v$ |
| max deg(u) | Maximum degree of neighboring nodes |
| min deg(u) | Minimum degree of neighboring nodes |
| mean deg(u) | Average degree of neighboring nodes |
| var deg(u) | Variance in the degrees of neighboring nodes |
| dominance | Diagonal dominance |
| decay | Diagonal decaying |
| pos | The position of the node in the matrix |

---

**Algorithm 2** Pseudo-code for NeuralIF preconditioner.

---

1: **Input:** Graph representation of the spd system of linear equations $\boldsymbol{Ax} = \boldsymbol{b}$.
2: **Output:** Lower-triangular sparse preconditioner for the linear system which is an incomplete factorization.
3: ▷ *NeuralIF preconditioner computation:*
4: Compute node features $\boldsymbol{x}_i$ shown in Table 4
5: Apply graph normalization
6: Split graph adjacency matrix into index set for the lower and upper triangular parts, $L$ and $U$.
7: **for** each message passing block $l$ in $0, 1, \ldots, N-1$ **do**
8:     ▷ *update using the lower-triangular matrix part*
9:     $z_{ij}^{(l+\frac{1}{2})} \leftarrow \phi_{\boldsymbol{\theta}_{z,1}^l}(\boldsymbol{z}_{ij}^{(l)}, \boldsymbol{x}_i^{(l)}, \boldsymbol{x}_j^{(l)})$ for all $(i,j) \in L$
10:     $m_i^{(l+\frac{1}{2})} \leftarrow \bigoplus_{j \in \mathcal{N}_L(i)}^{(1)} z_{ji}^{(l+\frac{1}{2})}$
11:     $\boldsymbol{x}_i^{(l+\frac{1}{2})} \leftarrow \psi_{\boldsymbol{\theta}_{e,1}^l}(\boldsymbol{x}_i^{(l)}, m_i^{(l+\frac{1}{2})})$
12:     ▷ *share the computed edge updates between the layers*
13:     $z_{ji}^{(l+\frac{1}{2})} \leftarrow z_{ij}^{(l+\frac{1}{2})}$ for all $(i,j) \in L$
14:     ▷ *update using the upper triangular matrix part*
15:     $z_{ji}^{(l+1)} \leftarrow \phi_{\theta_{z,2}^l}(z_{ji}^{(l+\frac{1}{2})}, x_j^{(l+\frac{1}{2})}, x_i^{(l+\frac{1}{2})})$ for all $(j,i) \in U$
16:     $m_i^{(l+1)} \leftarrow \bigoplus_{j \in \mathcal{N}_U(i)}^{(2)} z_{ji}^{(l+1)}$
17:     $x_i^{(l+1)} \leftarrow \psi_{\theta_{e,2}^l}(x_i^{(l)}, m_i^{(l+1)})$
18:     **if** not final layer in the network **then**
19:       ▷ *add skip connections*
20:       $z_{ij}^{(l+1)} \leftarrow [z_{ji}^{(l+1)}, a_{ji}]^\mathsf{T}$ for all $(j,i) \in U$
21:     **else**
22:       $z_{ij}^{(l+1)} \leftarrow z_{ji}^{(l+1)}$ for all $(j,i) \in U$
23: Apply $\sqrt{\exp(\cdot)}$-activation function to final edge embedding of diagonal matrix entries $z_{ii}^{(N)}$.
24: Return lower triangular matrix with elements $z_{ij}^{(N)}$ for $i \leq j$.

---

align the block with our objective, we only consider the edges in the lower triangular part of the matrix in the first layer (see lines 8–11 in Algorithm 2) while in the second layer of the block only edges from the upper triangular part of the matrix are used (lines 14–17). However, the same edge embedding for the two steps are used (line 13). In the final step of the message passing block, we introduce skip connections and concatenate the edge features in the current layer with the original matrix entries (lines 18–23). Note that, since the matrix $\boldsymbol{A}$ is symmetric it holds that $(i,j) \in L \Leftrightarrow (j,i) \in U$ where $U$ and $L$ are the index set corresponding to the non-zero elements in the upper and lower triangular matrix.

Here, we denote the updates after the first message passing layer with the superscript $(l + \frac{1}{2})$ to explicitly indicate that it is an intermediate update step. The final output from the block is then computed based on another message passing step which utilizes the computed intermediate embedding. Further, $\mathcal{N}_L(i)$ denotes the neighborhood of node $i$ with respect to the edges in the lower triangular part $\boldsymbol{L}$ and $\mathcal{N}_U(i)$ the ones from the upper triangular part $\boldsymbol{U}$ respectively which are utilized for the message passing. There are always at least the diagonal elements in the neighborhood of each node and therefore, the message computation is well defined.

**Network design** Both the parameterized edge update $\phi$ and the node update $\psi$ functions in every layer are implemented using two layer fully-connected neural networks. The inputs to the edge update network $\phi$ are formed by the node features of the corresponding edge and the edge features themselves leading to a total of $8 + 8 + 1 = 17$ inputs in the first message passing step and one additional input in later steps. The edge network outputs a scalar value. For each network, 8 hidden units are used and the `tanh` activation function is applied as a non-linearity. To compute the aggregation over the neighborhood $\oplus$, the `mean` and `sum` aggregation functions are used – respectively in the first and second step of the message passing block – which is applied component-wise to the set of neighborhood edge feature vectors. The node update function $\psi$ takes the aggregation of the incoming edge features as well as the current node feature as an input ($1 + 8 = 9$).

The NeuralIF model used in the experiments consists of 3 of the message passing blocks introduced in Section 3. Resulting in a total of six GNN message passing steps with two skip connections from the input matrix $A$. Graph normalization prior to the message passing leads to faster overall convergence of the training process (Cai et al., 2021). Note, that the graph normalization step, that is additionally applied to stabilize the training, is independent of the graph topology and only operates on the node feature vectors. The forward pass through the model is described in Algorithm 2.

**Network training**  We are training our model for a total of 50 epochs using a batch size of 5 for the synthetic problems and 1 for PDE problems. The Adam optimizer with initial learning rate 0.001 is used. Due to the small batch size and the loss landscape, we utilize gradient clipping to restrict the length of the allowed update steps and reduce the variance during stochastic gradient descent. While we use the Hutchinson's trace estimator with $m = 1$ during training which allows a full vectorized implementation of the problem loss function. We compute the full Frobenius norm during the validation phase to avoid overfitting and ensure convergence. Further, we use early stopping in order to avoid overfitting of the model based on the validation set performance by using the number of iterations the learned preconditioner takes on the validation data as the target.

## C.2  Baseline implementation

**Conjugate gradient method**  The conjugate gradient method is implemented both in the preconditioned form (as shown in Algorithm 1) and, as a baseline, without preconditioner using PyTorch only relying on matrix-vector products which can be computed efficiently in sparse format (Paszke et al., 2019). In order to ensure an efficient utilization of the computed preconditioners, the forward-backward substitution to solve for the search direction is implemented using the triangular solve method provided in the numml package for sparse linear algebra (Nytko et al., 2022). The conjugate gradient method is run until the normalized residual reaches a threshold of $10^{-6}$, see Section 3 for details.

**Baseline preconditioners**  The Jacobi preconditioner is due to its computationally simplicity, directly implemented in PyTorch and scipy using a fully vectorized implementation. It can be seen in the numerical experiments that the overhead for computing this method is very small compared to the more advanced preconditioners and does not effect the overall runtime significantly.

The incomplete Cholesky preconditioners and its modified variants with dynamic sparsity patterns and additional fill-ins are implemented using the highly efficient C++ implementation of ILU++ with provided bindings to python (Mayer, 2007; Hofreither, 2020). Due to the sequential nature of the computation, it is of critical importance that the utilized implementation is efficient.

The data-driven NeuralPCG baseline – which uses a simple encoder-decoder GNN architecture – (Li et al., 2023) is implemented in PyTorch and PyTorch Geometric following a very similar approach to NeuralIF. The hyperparameters for the model are chosen as specified in the paper: using 16 hidden units with a single hidden layer for encoder, decoder and GNN update functions and a total of 5 message passing steps. This results in a total of 10 177 parameters in the model which means the parameter count is over 5 times higher than our proposed architecture which makes inference using the NeuralPCG model more expensive as we observe in the experiments. Due to computational limits, the batch size is reduced during training compared to the original paper.

The results obtained from the model are similar to the one presented by Li et al. (2023). However, in our experiments we observed a higher pre-computation time which can be explained by the fact that different hardware acceleration is used. Further, in our experiments different sparsity patterns with more non-zero elements and larger matrices are used showing the poor computational scaling of the NeuralPCG model for large problem instances. In terms of resulting number of conjugate gradient iterations, our experiments match the results from the paper.

**Compute**  To avoid overhead due to the initialization of the CUDA environment, we run a model warm-up with a dummy input, before running inference tests with our model. The (preconditioned) conjugate gradient method – including the sparse forward-backward triangular solves for the utilized preconditioner – is run on a single-thread CPU. The learned preconditioner is computed using a single NVIDIA-Titan Xp GPU.

# D    Additional Results

Here, we present additional results and extensions of the baseline NeuralIF preconditioner presented in this paper. We both describe how to include reordering into the learned preconditioner as well as show more in-depth results. In Section D.2, we conduct an ablation study, probing the different loss functions proposed to train the learned preconditioner. Finally, we show the convergence behavior of the PCG algorithm for the different preconditioners and compare the performance with sparse direct methods.

## D.1    Matrix reordering

It is possible to directly combine our learned preconditioner with existing reordering methods. Classical reordering schemes such as COLMAD are executed in the symbolic phase before obtaining the values for the non-zero elements in the preconditioner (Gonzaga de Oliveira et al., 2018). Since our message passing scheme implicitly takes into account the ordering of the nodes, the nodes can simply be relabeled prior to running the forward pass of the neural network which influences the split into lower and upper triangular matrices that are used for the message passing. This ordering could also be learned alongside the values to fill in which is an interesting area for future research.

## D.2    Comparison of loss functions

In this section, we compare the different loss functions proposed to train the learned preconditioner. Computing the condition number as proposed in Sappl et al. (2019) as a loss function, is computationally infeasible in our approach since the underlying problems are large-scale compared to the problems encountered in the previous work. For the matrices used in our experiments, computing the condition number in the forward process is often computationally very expensive and therefore, the approach is omitted. We compare instead the full Frobenius norm as a loss as shown in objective (6a) and the stochastic approximation we suggest using Hutchinson's trace estimator as shown in equation (7).

Our loss is self-supervised and thus it is sufficient to have access to the problem data $\boldsymbol{A}_i, \boldsymbol{b}_i$ but not the solution vector $\boldsymbol{x}_i$. This saves significant time in the dataset generation. Further, it allows us to train a preconditioner on an existing dataset without solving all problems beforehand or even applying the learned preconditioner on the training data. The full Frobenius loss is also self-supervised but requires a matrix-matrix multiplication which leads to a large memory consumption and expensive forward and backward computation leading to longer training times. Further, matrix-vector multiplications are significantly easier to optimize and build a cornerstone of the CG algorithm allowing for example matrix-free implementations.

The stochastic and full Frobenius loss functions are compared in Table 5. We are training all models for 20 epochs and use a batch size of 1 on the synthetic dataset. Thus, the number of parameter updates in the training for all models is the same.    Using the full Frobenius norm as a loss function is significantly more computationally expensive compared to the stochastic approximation and, further, requires significantly more memory which means we can not apply it to large-scale problems. Training a model for 20 epochs with the full loss takes roughly 5 hours, while using the stochastic approximation or supervised loss, training is finished in around 30 minutes without decreasing the obtained preconditioner performance. Further, the

Table 5: Comparison of NeuralIF preconditioner performance on the synthetic problem set trained using different loss functions. The validation loss for both training loss functions is the Frobenius norm distance given in equation (6a). Time-per-epoch lists the training time for one epoch of training data (consisting of 1 000 samples), using a batch size of 1, in seconds.

| Loss function | Validation Loss ↓ | Time-per-epoch ↓ |
|---|---|---|
| Frobenius | **324.2** | 850 |
| Stochastic | 325.7 | **90** |

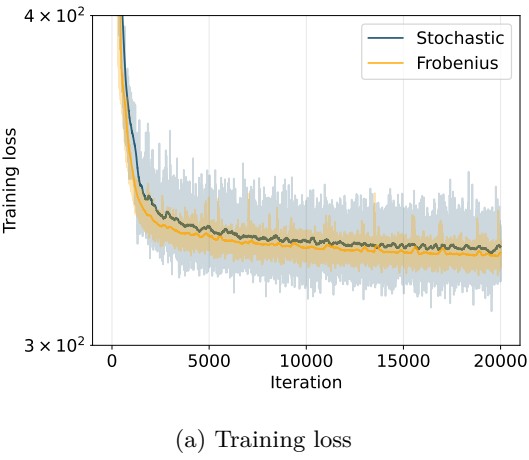

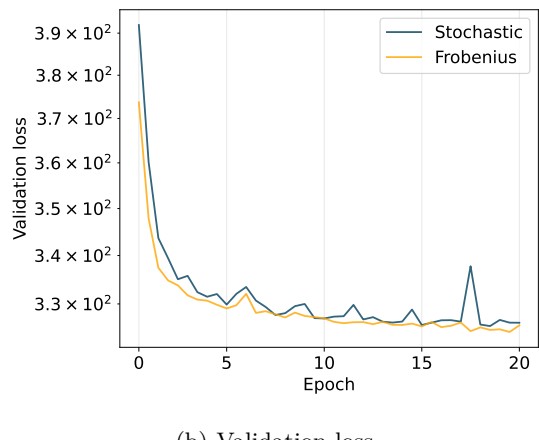

(a) Training loss

(b) Validation loss

Figure 7: Comparison of the training and validation loss for the Forbenius norm minimization and the stochastic approximation of the loss proposed by us. Here, we only compare the loss functions based on iteration count, not the actual time required to run the model training.

reduced memory consumption allows us to increase the batch size in other experiments leading to an even larger speedup in training time and additional variance reduction during the training process.

Even though we use a stochastic approximation to train our model, the validation loss computed as the full objective does not increase significantly. Both the training loss and the validation loss for the different loss functions are compared in Figure 7. Even though the variance for the stochastic approximation is significantly higher as expected, the validation performance of the two loss functions is nearly identical. One explanation for this is that the noise in the stochastic gradient descent method is already very large influencing the optimization procedure. Further, the added noise might help prevent overfitting on the training samples leading to a better performance of the stochastic loss in terms of validation loss.

### D.3 PCG convergence

Here, additional results for the synthetic dataset are shown. Notably, Figure 8 shows the convergence of the different (preconditioned) CG runs on a single problem instance from the synthetic dataset where both the residual – which is also used as a stopping criterion – and the usually unavailable distance between the true solution and the iterate are shown. Further, for each problem the worst-case bound depending on the condition number $\kappa(\boldsymbol{A})$ from equation (1) is displayed. We can see that the distance to the true solution is bounded by the theoretical $\kappa(\boldsymbol{A})$-bound and monotonically decreasing. The residual itself does not need to be decreasing and often increases in the first few iterations in our experiments.

The distribution of the eigenvalues of the linear equation system – shown in Figure 3 in the main paper for this problem instance – directly influence the convergence behavior of the method. However, as it is commonly observed in the CG method, the $\kappa(\boldsymbol{A})$-bound only describes the convergence behavior locally and overall, the algorithm converges significantly faster than the worst-case bound (Carson et al., 2023). We can, however, observe that the better conditioning still improves the convergence significantly and the obtained speedup is proportional to the speedup obtained in the worse-case bound making the condition number – at least for the problem instances considered in this dataset – a good measure for the performance of the conjugate gradient method. Further, we can see that our learned NeuralIF preconditioner is especially effective in the beginning of the CG algorithm and is able to outperform IC(0) for the first 100 iterations showing a significantly different convergence behaviour in this region.

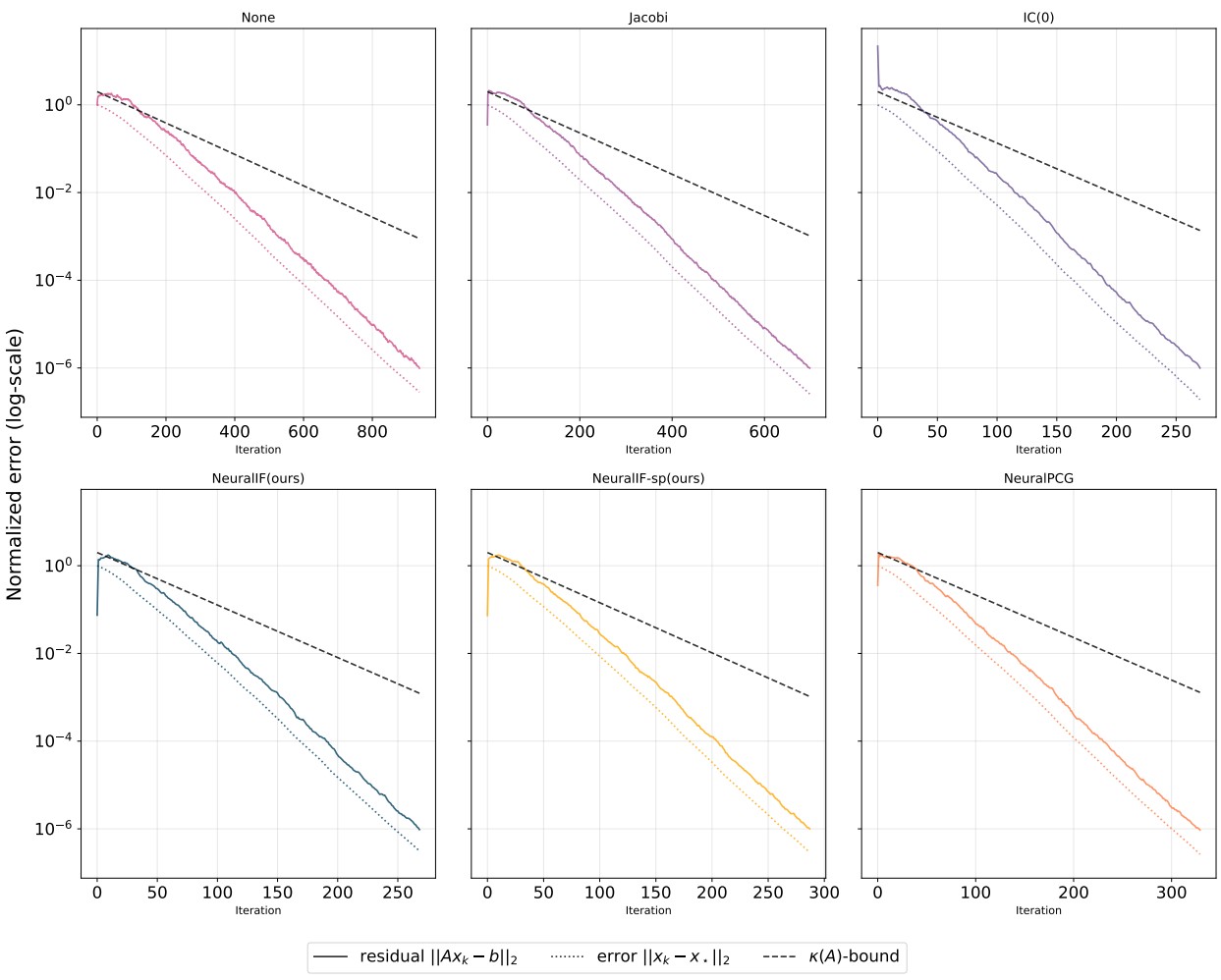

Figure 8: Convergence of the different preconditioned linear systems on a synthetic problem instance.

## D.4 Comparison with sparse direct solvers

Here, we compare the performance of the (preconditioned) conjugate gradient method with applying sparse direct solvers to the problem instances (Scott & Tůma, 2023). We are using the sparse Cholesky decomposition with GPU support of the CHOLMOD routine available from the SuiteSparse package with the Python interface provided by `cholespy`. We apply the supernodal factorization strategy and use nested dissection reordering to the matrix (Nicolet et al., 2021; Chen et al., 2008).

**Synthetic problems**   For the synthetic dataset the average time to solve each problem using the direct solver is 91 seconds which is significantly longer than the conjugate gradient based solvers presented in Table 1. This is due to the fact that no structured sparsity in the problems is present that can be efficiently exploited during the symbolic-solving phase due to the construction of the problems. Therefore, using iterative methods is far superior in this case even though the performance of the CG method is not tuned.

**Poisson problems**   In contrast, problems arising from the discretization using the finite-element method are well known to be efficiently solvable using sparse direct methods due to their inherent sparsity structure. However, by taking domain knowledge into account it is possible to accelerate the CG solver further using the popular multigrid methods which constructs the preconditioner based on the properties of the problem domain as previously discussed.

The average time to solve the problems from the small Poisson dataset is 0.45 seconds which is slightly faster than the preconditioners without any additional fill-ins but on par with the additional results shown in Table 2. For the larger problem instances, the direct method scales slightly better in our experiments requiring an average time of 2.7 seconds to solve the problem instances. In comparison, the best solver using the conjugate gradient method requires an average of 4.4 seconds (IC) and 4.5 seconds (NeuralIF) respectively over all problem instances. Thus, overall the difference between the two implementations is not huge but sparse direct solvers using GPU show a slightly better scaling in our experiments.

Note, however, that the numerical comparison presented here between the different methods is in favor of the sparse direct solvers. While different preconditioner are evaluated using the same solver implementation, it makes sense to compare the timing between the different preconditioning techniques even though the solver itself is not optimized for performance. Further, as described in Section D.1 additional techniques such as reordering of rows and columns can be applied to obtain better preconditioners by applying additional heuristics. These techniques are already integrated into the sparse direct solvers applied here. The full implementation of the sparse direct solver is optimized in a low-level language while our CG method is executed in Python, leading to slower solving times.

