# OpenReview forum: "Neural incomplete factorization: learning preconditioners for the conjugate gradient method"
_TMLR — Accepted by TMLR_

### Review · Reviewer_WP76 · 2024-06-13

**Summary Of Contributions:**

The paper proposes a new way of using graph neural networks to learn a preconditioning matrix for efficiently solving large-scale systems of linear equations using conjugate gradients. The proposed method uses a graph neural network to learn to generate an incomplete factorization of the system matrix. Evaluation of the proposed method is performed on synthetic problems as well as on real-world ones (problems obtained through discretized PDEs).

**Audience:**

Yes

**Broader Impact Concerns:**

No concerns in this regard.

**Claims And Evidence:**

Yes

**Requested Changes:**

As I was not able to find any clear errors in the paper, I do not have any requested changes.

I only have one comment that might be useful:
At the end of the first paragraph of section 2.1, it is said that Algorithm 1 illustrates the conjugate gradient method. However, my impression is that this algorithm illustrates the “preconditioned version” of the conjugate gradient method. Perhaps it would be good to clarify this somehow.

**Strengths And Weaknesses:**

Unfortunately, I am not an expert in the topic of this paper and my knowledge about it is rather limited. I am therefore afraid that I was not able to holistically judge the importance, novelty and correctness of the paper.

However, I certainly enjoyed reading the paper. I think the paper is very well written. Even for someone not directly familiar with the topic (like me), it is possible to follow and understand the paper.
I was further not able to find any clear errors in the paper, or other reasons why this paper should not be accepted. My impression is that the main claims of the paper are consistent with the provided results.

I therefore do not see objections to accepting this paper at TMLR (provided other reviewers more familiar with this topic do not raise important concerns). I certainly learned a lot from reading this paper.

---

> ### Author Response · Authors · 2024-06-21
>
> We would like to thank the reviewer for their careful reading of the paper and the positive evaluation.
>
> We adjusted the mentioned paragraph and clarified that as correctly observed, Algorithm 1 shows the preconditioned version of the conjugate gradient method. From this the original algorithm can be recovered by setting P=I.
>
> We are happy to answer any additional questions that might arise.

---

### Review · Reviewer_iah6 · 2024-06-15

**Summary Of Contributions:**

The paper proposes to use a graph neural network to learn a lower-triangular matrix $L$ that can be used to approximate the Cholesky factorization of a given input matrix. The output $L$ is then used as to precondition the conjugate gradient method.

**Audience:**

Yes

**Broader Impact Concerns:**

N.A.

**Claims And Evidence:**

Yes

**Requested Changes:**

N.A.

**Strengths And Weaknesses:**

The paper is clearly written.

The weaknesses in my humble opinion are as follows:
- Experimental comparisons are unfair. The time that outputs a preconditioner is measured on different devices (GPU and CPU). To me, it only makes sense if everything is implemented in GPU (or CPU). Specifically, it should be very easy to run GNNs on CPU, given the GPU implementations.
- The proposed method does not show promising experimental results compared to prior works. Moreover, the paper seems to miss several relevant papers which can solve very large scale linear equations using randomized numerical linear algebra. See, e.g., some papers of Joel Tropp. For example: (https://epubs.siam.org/doi/abs/10.1137/21M1466244).
- The proposed objective in Eqs. 6a and 6b appear to be a *contradiction* to me. The contradiction is this. The paper aims to solve linear equations. To do so, the paper considers Eqs 6a and 6b, which are harder problems than solving linear equations. Why does this even make sense? If GNNs were able to solve Eqs. 6a and 6b, I would argue that it could also directly output a solution to the linear equations more efficiently (as the linear problem is easier). How do the authors even see this?
The other problem involves the training cost of GNNs. The paper didn't report the cost numerically, while it discussed how training GNNs could be a major limitation, and in light of this, it is unclear why this line of research (L2P) is of interest. I am not a fan of L2P. In my eyes, the paper tackles an interesting problem with the wrong tools, efforts have been put in, but there is no way to fix that.

---

> ### Author Response · Authors · 2024-06-21
>
> We would like to thank the reviewer for their careful reading and feedback of our manuscript. We will address each point individually and hope to clarify some misconceptions about our method.
>
> > Experimental comparisons are unfair. The time that outputs a preconditioner is measured on different devices (GPU and CPU). To me, it only makes sense if everything is implemented in GPU (or CPU). Specifically, it should be very easy to run GNNs on CPU, given the GPU implementations.
>
> One key advantage of our method is that the neural network based approach can be efficiently and easily run using a GPU. In contrast, the classical incomplete Cholesky method can not be run efficiently on a GPU due due its sequential computational nature. In our opinion, each algorithm should be run on the hardware which best supports the required computations rather than forcing them to be executed on the same devices.
>
> We could run our method on CPU which would give the highly customized implementation of incomplete Cholesky at an advantage. On the other hand, we could run incomplete Cholesky directly on GPU. However, this would increase the P-time by orders of magnitude since the computations are not tailored for the GPU architecture. Instead, we decided to compare the methods using the hardware devices that best support the required computations for each algorithm.
>
> > The proposed method does not show promising experimental results compared to prior works.
>
> We would like to ask the reviewer to clarify which prior work we should compare our method to in order to obtain more convincing results.
>
> Further, we want to emphasize that we do not claim to have the best overall preconditioner but rather develop a method competitive with algebraic incomplete factorizations. We compare our results to the previously proposed data-driven NeuralPCG model which we clearly outperform both in time to compute the preconditioner and in terms of solving time of the linear equation system as well as standard incomplete factorization methods.
>
> > Moreover, the paper seems to miss several relevant papers which can solve very large scale linear equations using randomized numerical linear algebra. See, e.g., some papers of Joel Tropp. For example: ([https://epubs.siam.org/doi/abs/10.1137/21M1466244](https://epubs.siam.org/doi/abs/10.1137/21M1466244)).
>
> The Nyström preconditioner differs in two fundamental assumptions from our problem formulation:
> - As shown in equation (1.1) of the suggested paper, the problem is assumed to be a sum of a semi-positive definite matrix and a scaled identity matrix. Further, it is assumed that the first matrix can be efficiently approximated using a low-rank approximation.
> - The problem matrix when considering Nyström preconditioner is in general non-sparse and neither is the constructed preconditioner.
>
> In contrast, we only make the assumption that the matrix is spd and very sparse. Assuming additional structure makes the problem significantly different and, therefore, different tools are required to solve them. Problems arising in the paper mentioned above arise especially in the solution to regularized least-squares problems while our method is tailored to be applied to problems in numerical optimization such as the example using PDE discretizations shows.
>
> > The proposed objective in Eqs. 6a and 6b appear to be a _contradiction_ to me. The contradiction is this. The paper aims to solve linear equations. To do so, the paper considers Eqs 6a and 6b, which are harder problems than solving linear equations. Why does this even make sense?
>
> Equation 6 is only used as a training objective. In other words, we use equation 6 to optimize the parameters of the GNN (which is in itself a non-convex problem) denoted by $\theta$ . However, the solution of the optimization problem with respect to the input $A$ is never explicitly computed but only approximated through the neural network output. This is a classical scheme in learning-to-optimize approaches where the loss function used to train the network can differ from the objective function which is aimed to be minimized in the downstream task. Our neural network predicts the solution for every $A$ then but no guarantees about this output can be given annd we do not expect our neural network to solve this problem exactly. However, the numerical experiments show that the constructed output performs well as a preconditioner for the system.

---

> ### Author Response · Authors · 2024-06-21
>
> > If GNNs were able to solve Eqs. 6a and 6b, I would argue that it could also directly output a solution to the linear equations more efficiently (as the linear problem is easier). How do the authors even see this?
>
> Previous approaches have tried to directly approximate the solution of the system [1], as mentioned in the related work section. The main problem with using the GNNs directly to predict the solution of the linear equation system is that no convergence guarantees about the method can be given in this case. Further, a fully data-driven method can only obtain at most a few digits of accuracy which is not sufficient for a lot of applications. In contrast, our method aims to combine the advantages of these technologies by learning the preconditioner which allows us to accelerate the solution time while maintaining the strong convergence guarantees of the conjugate gradient method.
>
> [1] Grementieri and Galeone, Towards Neural Sparse Linear Solvers, Preprint arXiv:2203.06944, 2022
>
> > The other problem involves the training cost of GNNs. The paper didn't report the cost numerically, while it discussed how training GNNs could be a major limitation, and in light of this, it is unclear why this line of research (L2P) is of interest. I am not a fan of L2P. In my eyes, the paper tackles an interesting problem with the wrong tools, efforts have been put in, but there is no way to fix that.
>
> We only compare the training time currently in the appendix but we agree that it is more suitable to include this information in the main text which weupdated this in our revised manuscript and included it in the result section. However, note that once the network is trained once no additional cost during inference is needed other than the reported P-time.
>
> We would like to ask the reviewer for clarification about what they mean with L2P. Our approach is grounded in the learning-2-optimize (L2O) scheme while L2P usually refers to learning-to-prompt, a continual learning scheme, which focuses on adjusting the model parameters for new data samples where typically the data distribution is changing over time.
>
> We are happy to answer and follow up questions and hope to have clarified some of the misconceptions about our proposed method.

---

> ### Author Response · Authors · 2024-07-01
>
> Dear Reviewer, as the discussion period is coming to an end soon, we wanted to reach out once more to see if you have any further thoughts or feedback based on our recent rebuttal. We highly value your insights and would appreciate any additional comments or questions you might have.

---

### Review · Reviewer_oPBk · 2024-06-18

**Summary Of Contributions:**

The authors propose NeuralIF (Neural Incomplete Factorization), a GNN (Graph Neural Network) architecture for computing preconditioners for solving systems of linear equations involving large, symmetric, positive-definite (SPD) sparse matrices. In particular, the Coates graph representation of the lower triangular part of the sparse matrix is considered as input to the GNN. This graph drives a pair of novel message passing substeps which update node and edge features, respectively initialized to a vector of numbers mostly capturing degree and diagonal dominance / decaying information and the input matrix value. Then the resulting scalar edge encodings are utilized as the entries in the Cholesky decomposition factors of returned computed preconditioner. During model training, loss terms representing sparsity (optional) and the "reconstruction" loss of the input matrix from Cholesky factors (Frobenius norm), the latter approximated via Hutchinson trace estimation. Numeric experiments over both synthetic and Poisson-PDE-generated matrix instances of sizes ranging from tens to a few hundreds of thousands and matrix sparsities of the order of 1% (or less  for larger matrices) are presented. NeuralIF is fast in generating "high-quality" preconditioners (i.e. fewer iteration steps in the subsequent preconditioned CG (conjugate gradient) for solving the associated linear equations' system; decrease in the condition number) as compared to classic (incomplete Cholesky (IC) and variants) and  neural approaches (NeuralPCG), additional possessing some attractive properties including breakdown avoidance (in classical methods) and generalization (across matrix training sets and their scales).

**Audience:**

Yes

**Claims And Evidence:**

Yes

**Requested Changes:**

- Hutchinson trace estimation in the loss function seems to be using only $m=1$ samples during training (B.1 in the Appendix). If this is the case, then the quality of the approximation is expected to be extremely poor. However, resulting preconditioners are really competitive in quality with traditional, heavily-tested and well-grounded in linear algebra methods. Is there an intuition, why still NeuralIF performs that well?

- In Table 1, there is definitely a performance edge of NeuralIF (with respect to IC) when generating a "high-quality" preconditioner. However, this edge is not very pronounced, in the sense that a small decrease in its quality could result in longer solution times overall (and end-to-end times since generation (P-time) is not the dominant part in the timings reported for this range of parameters). A small ablation study (e.g. different sparsities, matrix sizes and number of samples in approximating the trace for the synthetic part) would certainly strengthen the validity of the central claim in the paper that NeuralIF can be faster thus more attractive than competition baselines (particularly IC).

- The manuscript leverages Coates graph representation for matrices. It is not clear how this is connected with the pair of messaging substeps in their proposed GNN architecture. Although they do mention that matrix multiplication is related to concatenating graph representations, this view seems to be linked to Koenig (rather than Coates) graph representation. Overall, a clean outline of the motivation of their two substeps in the GNN from their graph view of the input matrix would be very beneficial for the reader seeking to appreciate the intuition underlying the approach.

- On an ablation/intuition theme (could add to revealing the relative importance of design choices): Are the results expected to be sensitive to a different mix of initial node features? For example if we zero only  the node-degree-related or only the diagonal-dominance/decaying-related initializing features, how the quality of the preconditioner (e.g. the condition number) will change?

**Strengths And Weaknesses:**

**Strengths**
- Lightweight model, few parameters.
- Fast generation of "high-quality" preconditioners.
- Applicable to matrices of different sizes without retraining; generalizes and avoids breakdowns necessitating expensive restarts as in some of the baselines.
- Less variability in the time to generate the preconditioner.


**Weaknesses**
- A trained model is needed which in turn implies a set of training matrices. Then questions like how to amortize the cost of training or how to make sure that the training distribution would give a model that would generate performant preconditioners for the target matrices (generalization) arise. These are not present in traditional approaches, although then, as nicely mentioned in the limitations, manual tuning would be needed instead. These are issues of L2O (Learning to Optimize) frameworks in general, but still limit the practicality envelope of the approach.
- The preferable platform includes both a CPU and a GPU. Although CPU-only training, inference and solving would be feasible (thanks to the small size of their GNN), going CPU-only (as in traditional baselines) would change total computation times when using NeuralIF for solving a problem. But then it is not clear how this setup, would change the timing comparison ranking, which is one key benefit of NeuralIF; especially if the number of systems to solve is limited and training time (along the lines of Table 5 in Appendix C.2) is also taken into account.

---

> ### Author Response · Authors · 2024-06-21
>
> We would like to thank the reviewer for their careful reading of our manuscript and their helpful comments. We will address each point individually and updated the manuscript to incorporate suggested changes.
>
> > Hutchinson trace estimation in the loss function seems to be using only $m=1$ samples during training (B.1 in the Appendix). If this is the case, then the quality of the approximation is expected to be extremely poor. However, resulting preconditioners are really competitive in quality with traditional, heavily-tested and well-grounded in linear algebra methods. Is there an intuition, why still NeuralIF performs that well?
>
> As the reviewer correctly observes we only use a single sample for the Hutchinson's trace estimator since this allows to compute the loss very efficiently. Despite leading to a good preconditioner, we also observe, that using the loss approximation we obtain very similar results in terms of validation loss (which we always compute using the full frobenius norm) as shown in Figure 7 so the choice of the sampling size leads to the minimization of the desired loss function as well as an efficient preconditioner.
>
> One possible explanation for the good performance despite the poor approximation could be the small overall number of parameters of our model in comparison to the high output dimension. Further, since we are using a stochastic gradient descent based optimization scheme, the loss computed to train the model is very noisy anyway due to the small batch size. However, the Adam optimizer is known to show good results in these scenarios by automatically adjusting the learning rate.
>
> > In Table 1, there is definitely a performance edge of NeuralIF (with respect to IC) when generating a "high-quality" preconditioner. However, this edge is not very pronounced, in the sense that a small decrease in its quality could result in longer solution times overall (and end-to-end times since generation (P-time) is not the dominant part in the timings reported for this range of parameters). A small ablation study (e.g. different sparsities, matrix sizes and number of samples in approximating the trace for the synthetic part) would certainly strengthen the validity of the central claim in the paper that NeuralIF can be faster thus more attractive than competition baselines (particularly IC).
>
> Generating good problems that allow an appropiate ablation study in this context is challenging. This is due to the fact that the matrix properties one would like to examine are correlated. For example, when decreasing the sparsity, the problem becomes more and more similar to the identity matrix which is often trivial to solve. In general, when creating diverse problem instances to analyse the properties of the methods these effects should be taken into account using tools from random matrix theory.
>
> We think two seperate questions related to this need to be answered in the future
> - How does a trained model generalize to out of distribution data?
> - How diverse should the training data be (ie how much variance should be present in the training data) to still obtain well-functioning data-driven preconditioner
>
> However, these are not real ablation studies but rather assesing the generalization performance of the learned model.
>
> Regarding the first question, we do some analysis with regard to problems arising from the PDE problems as a natural generalization of the problems exist by using finer mesh-sizes. For the synthetic problems, we instead focused on a more narrow class of problems in order to keep all problems from the distribution at a similar level of comparison. However, these problems share less algebraic structure with each other so generating a "natural extension" of these problems to larger or sparser problems is not straightforward.
>
> The results of NeuralIF-sp show, however, that our model is quite robust to changes in terms of the output and many good preconditioner can be found by the model. This can be seen by the fact that the model only uses 50% of the non-zero elements of $A$ but still reduces the number of required iterations drastically. This shows that the quality of the output is relatively robust with respect to the input of the network.
>
> We do not attempt to answer the second question as this requires a very careful dataset generation to draw general conclusions as described above.

---

> ### Author Response · Authors · 2024-06-21
>
> > The manuscript leverages Coates graph representation for matrices. It is not clear how this is connected with the pair of messaging substeps in their proposed GNN architecture. Although they do mention that matrix multiplication is related to concatenating graph representations, this view seems to be linked to Koenig (rather than Coates) graph representation. Overall, a clean outline of the motivation of their two substeps in the GNN from their graph view of the input matrix would be very beneficial for the reader seeking to appreciate the intuition underlying the approach.
>
> We would like to thank the reviewer for this useful comment. We agree indeed with the observation that the current description of the underlying graph used for the message-passing scheme lacked clarity in the original draft.
>
> We consider the Coates graph of the lower triangular matrix for the message passing but look at the unrolled message passing steps and the concatenation of matrices correspond to the Koenig graph representation. We think this might have caused some confusion in the current draft. The most important concept of the message passing substeps is the fact that they contain a positional encoding of the nodes in the graph which is otherwise implicitly assumed when processing the output of the GNN.
>
> In the updated manuscript we clarified this relationship and added an additional figure in order to visualize better the two separate message passing steps when we unroll the GNN. We hope these changes make the method more understandable for the reader.
>
> > On an ablation/intuition theme (could add to revealing the relative importance of design choices): Are the results expected to be sensitive to a different mix of initial node features? For example if we zero only the node-degree-related or only the diagonal-dominance/decaying-related initializing features, how the quality of the preconditioner (e.g. the condition number) will change?
>
> We only conducted a minimal level of hyper-parameter tuning when developing our model so we expect the performance to not change dramatically when using different initial features. We utilize the features previously introduced in the literature which are easily to compute leading to a low overhead. The main reason for choosing additional node features is to obtain a higher dimensional embedding which allows us to parameterize the message passing steps using more expressive neural networks. Therefore, we do not expect the results to be very sensitive to the chosen set of node features.
>
> We are happy to answer any additional questions that might arise.

---

> ### Author Response · Authors · 2024-07-01
>
> Dear Reviewer, as the discussion period is coming to an end soon, we wanted to reach out once more to see if you have any further thoughts or feedback based on our recent rebuttal. We highly value your insights and would appreciate any additional comments or questions you might have.

---

### Review · Reviewer_ym4F · 2024-06-24

**Summary Of Contributions:**

The authors introduce NeuralIF which learns the precondition by using a message-passing algorithm on a graph. This method is shown to be fast and efficient when the linear system is sparse. This method avoids matrix-matrix multiplications by learning an orthonormal basis under A-norm which reduces the computational complexity to matrix-vector multiplications.

**Audience:**

Yes

**Claims And Evidence:**

Yes

**Requested Changes:**

NA

**Strengths And Weaknesses:**

Good expositions
Experiments are well organized and comprehensive
Background and literature is well presented

Unfortunately, I have little to no idea about this field and previous work hence I am likely missing some strengths and weaknesses on the technical aspects.

---

> ### Author Response · Authors · 2024-07-01
>
> We would like to thank the reviewer for their careful reading and positive evaluation of our manuscript. We are happy to answer any additional questions that might arise.

---

### Review · Reviewer_57D1 · 2024-08-19

**Summary Of Contributions:**

This paper considers using GNN to produce preconditioners for solving linear equation using Preconditioned Conjugate Gradient (PCG) method. The authors propose a novel unsupervised method for learning preconditioners by proposing a GNN model and a loss function. THe utility of the method is demonstrated in experiments that compare it to classical preconditioning techniques (Jacobi, Incomplete Cholesky) and previous data-driven neural preconditioners (NeuralPCG).

**Audience:**

Yes

**Claims And Evidence:**

Yes

**Requested Changes:**

Experiments:
- To make a really convincing argument, I think the authors should also include experiments with matrices from Tim Davis's matrix collection (https://sparse.tamu.edu/about). The matrices are divided into types, so one possibility is to use small matrices of a type to learn the GNN and then apply them to larger matrices from the same type. That said, other setups can be thought of. The point is, that trying the algorithm on real world problems from the wild will make a much stronger claim.
- The experiments give focus only on small problems. More focus should be given to much larger problems. If feasible, conduct additional experiments on larger problems.
- Please also report the training time. (I assume that "Total time" does not include this). While I can accept the claim that training replaces manual development of a preconditioning method, I still think the reader should be reported the time it takes to prepare the preconditioner.

Complexity discussion:
Please report the complexity of computing the precondiioner *after* training (that is, to apply the model on a test problem). If possible, compare it the cost of Incomplete Cholesky.

You use the term "fill-in" but do not explain what it is, and how it is critical to the cost of computing IC and how well it approximates the matrix.

Please copy edit the manuscript. The English is at some places awkward .

- Sec 2.1 - the vectors are orthogonal in the A-inner product, not A-norm. Please make sure to use correct terminology.
- Page 2 - "Further, a large condition number needs not imply a slow convergence of the iterative scheme" - I would add the word "always", and say that small condition number does imply fast convergence.
- Page 3 - Paragraph starting with "Finding new preconditioners is an active research area but is often done on a case-by-case basis" makes a too strong claim, and not an accurate ones. It is true that the best precondioners are tailor-made, but there are general purpose methods like IC, support preconditioners, and algebraic multigrid.
- Page 4 "node n_i" -> "node i"?
- I don't think the term SPD has been defined. I think you mean "symmetric positive definite"?
- How is the objective in (6) related to the one used in Sparse Approximate Inverse Preconditioners?
- Discussion after (7) - note that after the square root the estimator is no longer unbiased! BTW, you might be able to compute gradients of the real objective by sampling w's and taking gradients with respect to them. Didn't check the details if this works.
- Last paragraph of page 5 and first paragraph of page 6 are very hard to understand. Please revise, and explain in more details.
- The use of the factor 1/2 in (8) is very heuristic and mysterious. I wonder how this is fine-tuned to the problems in the experiments...
- Discussion in "additional fill-ins and dropping" - how is the optimization done? adding all edges? Wouldn't that be expensive?

**Strengths And Weaknesses:**

Strengths:
- A novel method for learning preconditioners. The method is very elegant.
- The method is unsupervised.
- Experiments demonstrate the utility of the method.

Weaknesses:
- Experimental section is a bit weak:  (this item is the reason I checked "No" on "Claims And Evidence", though it is fixable).
    * Conducted only on a very small scale. On the scale of conducted experiments, direct methods are much superior.
    * Poisson PDE problem for a real-world problem is nice, but I think additional real-world experiments can be conducted (see "Requested Changes" section), and that can make the paper much stronger.
-  I am dubious about how well the method will work on large problems. In fact, the experiments reported by the authors show advantage mainly for small problems. The most the can say on Figure 4 (larger scale experiments) is that the results are on par with IC.

Overall, I am positive about the paper. The method is likely not that useful in practice, but it is a good advancement in this area, and a publishable contribution.

---

> ### Author Response · Authors · 2024-08-27
>
> We would like to thank the reviewer for the careful reading, helpful suggestions, and the overall positive evaluation of our paper.
>
> > To make a really convincing argument, I think the authors should also include experiments with matrices from Tim Davis's matrix collection.
>
> While we see the value of such experiments, the problem is that there are too few matrices in the collection that share a common structure that can be exploited by a learning-to-optimize approach such as ours. Specifically, the dataset only contains around 200 matrices that are spd and real, and only a handful come from the same application.
>
> As we discuss in the limitation section, a fundamental assumption of learning-to-optimize approaches is access to a large set of problems sharing some similarity. We thus see it as important future research directions to (i) create a good suitable dataset for learned preconditioners, and (ii) extending our method to make it possible to train with only a few matrices.
>
> > In fact, the experiments reported by the authors show advantage mainly for small problems. The most the can say on Figure 4 (larger scale experiments) is that the results are on par with IC.
>
> Indeed, our main claim is that "Our data-driven method exhibits state-of-the-art results for data-driven preconditioners and performs on par with the baseline incomplete factorization methods."
>
> > Conducted only on a very small scale. On the scale of conducted experiments, direct methods are much superior.
>
> It is not correct that direct methods are much superior in our experiments:
>
> *Dense direct solvers* are applicable to problems of this size but require more time (~6 seconds using a Cholesky based solver). Our experiments show that the conjugate gradient method gives a 10x sppedThis justifies the use of iterative solvers for these problem sizes as a speedup of 10x is achieved by using the conjugate gradient method in our experiments.
>
> *Sparse direct methods* perform even worse on the synthetic sparse matrices. For instance, using the SuperLU backend, the symbolic phase of the solver requires over 70 seconds to solve a single problem. (We hypothesise this is because the sparsity pattern lacks regular structure and therefore requires significant fill-in.)
>
> On the PDE problems, dense solvers can not be utilized due to the large problem size. Sparse direct solvers are competitive in our setting up to problems with 1 million non-zero elements. However, we also consider larger problems than this during our evaluation where sparse solvers are superior.
>
> Furthermore, we expect CG to compare even better in practice, since the numbers we report are based on our Python implementation of CG whereas the direct methods are implemented significantly more efficient.
>
> Compared to previous papers on learned preconditioners (NeuralPCG), the matrices in our experiments are 20x bigger. Further, 90% of matrices in the SuiteSparse dataset are of the same size as our test data showing that the considered scale of problems already covers and exceeds previous approaches in the literature.
>
> > Please also report the training time. (I assume that "Total time" does not include this). While I can accept the claim that training replaces manual development of a preconditioning method, I still think the reader should be reported the time it takes to prepare the preconditioner.
>
> The training time is included after the feedback from another reviewer in Section 4. This includes, however, the time required to run evaluation on the validation data where multiple systems of equations are solved as we state in the text.
>
> > Complexity discussion: Please report the complexity of computing the precondiioner after training (that is, to apply the model on a test problem). If possible, compare it the cost of Incomplete Cholesky.
>
> We added a discussion on the space and time complexity of the forward pass to the *model inference* section in our revised draft.
>
> > You use the term "fill-in" but do not explain what it is, and how it is critical to the cost of computing IC and how well it approximates the matrix.
>
> We discuss the term "fill-in" briefly in the section 2.1 in the paragraph about preconditioning. We add additional clarification about the importance of the parameter in the revised manuscript.

---

> > ### Comment · Reviewer_57D1 · 2024-09-01
> >
> > Regarding comparison to direct methods:
> > Obviously, for synthetic problems, where the sparsity structure is random, then direct methods are inferior, and will not work well. However, these are very far from real world problems. Further, I don't think IC and your method will work well for these. A good baseline for this problem is unpreconditioned CG.
> >
> > For the Poisson problem: not all your problems are larger then 1M non-zeros.. I do not see any reason not to report running time, even if they should that direct methods win. Furthermore, for a fair comparison you should use a direct method that uses parallel computing, preferably GPU acceleration. For example, instead of vanilla SuperLU, use SuperLU_DIST which has GPU support. Or, use CHOLMOD (part of SuiteSparse, https://github.com/DrTimothyAldenDavis/SuiteSparse), which also has GPU support.

---

> > > ### Author Response · Authors · 2024-09-02
> > >
> > > For synthetic problems, we do compare to the suggested unpreconditioned CG baseline in Table 1 (Preconditioner = None). As the results show both IC and our method clearly outperform this method.
> > >
> > > We would be happy to include a comparison to direct methods (using GPU support to ensure a fair comparison as rightfully suggested by the reviewer) in the final version of our submission.

---

> ### Author Response · Authors · 2024-08-27
>
> We would like to thank the reviewer for the helpfull comments on the manuscript. We adjusted the writing based on the feedback
>
> > Page 3 - Paragraph starting with "Finding new preconditioners is an active research area but is often done on a case-by-case basis" makes a too strong claim, and not an accurate ones. It is true that the best precondioners are tailor-made, but there are general purpose methods like IC, support preconditioners, and algebraic multigrid.
>
> We agree with the reviewer and adjusted the description accordingly.
>
> > I don't think the term SPD has been defined. I think you mean "symmetric positive definite"?
>
> We define the term in the first paragraph of section 2.1 where we use the term for the first time. The definition of spd matrices is included in the appendix.
>
> > How is the objective in (6) related to the one used in Sparse Approximate Inverse Preconditioners?
>
> We added a discussion on how our approach relates to SPAI preconditioners in the appendix.
>
> > Discussion after (7) - note that after the square root the estimator is no longer unbiased!
>
> Thank you for the insightfull observation. You are correct, however, the derived expression is an unbiased estimator for the squared loss function. Optimizing the parameters over the squared loss is equivalent to the original problem formulation and we changed the corresponding notatin in equation (6).
>
> > Last paragraph of page 5 and first paragraph of page 6 are very hard to understand. Please revise, and explain in more details.
>
> Thank you, we have revised the paragraph to make it more accessible, please see the updated manuscript.
>
> > The use of the factor 1/2 in (8) is very heuristic and mysterious. I wonder how this is fine-tuned to the problems in the experiments...
>
> This term is added to enhance numerical stability, similar to how the incomplete Cholesky method takes the square root of the diagonal elements. The reason behind this is that the diagonal elements get multiplied with each other in the loss function. Due to the exponential nature of the diagonal activation function this might cause numerical issues especially in the beginning of the training process. Therefore, we take the square-root of the elements, which is equivalent to dividing the values with 1/2 before applying the activation function.
>
> > Discussion in "additional fill-ins and dropping" - how is the optimization done? adding all edges? Wouldn't that be expensive?
>
> As pointed out by the reviewer adding all edges to the graph is computationally intractable. Instead, we use heuristics similar to incomplete Cholesky with level based fill ins such as using the sparsity pattern of $A^2$ as mentioned. We clarify this in the updated draft.

---

### Decision · Action_Editor_osWA · 2024-09-06

**Recommendation:** Accept with minor revision

**Comment:**

The proposed method is novel, and as per the reviews "very elegant", fast and high-rnough quality.  The training method is unsupervised, and hence the approach scales. The preconditioner itself is based on a Lightweight model with few parameters. It is also applicable to matrices of different sizes without retraining; generalizes and avoids breakdowns necessitating expensive restarts as in some of the baselines. Finally, experiments demonstrate the utility of the method though reviewers would have liked to see more convincing and comprehensive experiments at a larger scale.

**Audience:**

Yes, this paper is at the intersection of classical numerical methods and deep learning.

**Claims And Evidence:**

The paper develops the idea of training GNNs to serve as preconditioners for accelerating solving linear equations with PCG methods. This approach shows promise when compared to classical preconditioning techniques (Jacobi, Incomplete Cholesky) and previous data-driven neural preconditioners (NeuralPCG). While experiments could be strengthened to make a more convincing case for larger scale problems, the reviewers agree that the current contribution is publishable.